# LESS GAUSSIANS, TEXTURE MORE: 4K FEED-FORWARD TEXTURED SPLATTING

**Yixing Lao**[1,2†]  **Xuyang Bai**[2]  **Xiaoyang Wu**[1]  **Nuoyuan Yan**[2]  **Zixin Luo**[2]  **Tian Fang**[2]
**Jean-Daniel Nahmias**[2]  **Yanghai Tsin**[2]  **Shiwei Li**[2‡]  **Hengshuang Zhao**[1]

[1]HKU  [2]Apple

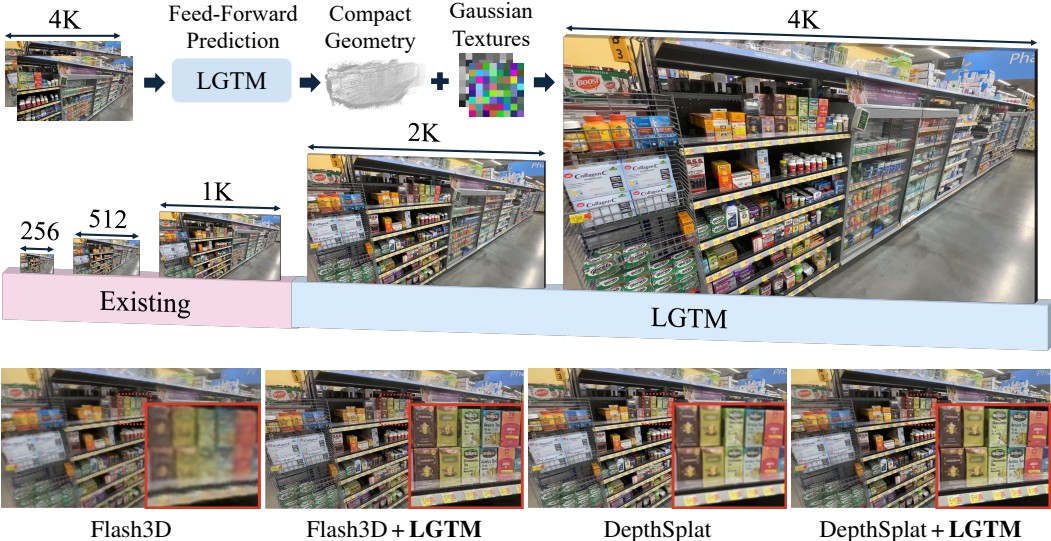

Figure 1: **LGTM enables feed-forward 4K textured Gaussian splatting.** LGTM enables high-fidelity 4K novel view synthesis, a capability previously intractable for feed-forward methods. By predicting compact Gaussian primitives paired with per-primitive textures, LGTM overcomes the resolution limitations of prior work, enabling high-quality reconstruction with significantly fewer primitives and no per-scene optimization.

## ABSTRACT

Existing feed-forward 3D Gaussian Splatting methods predict pixel-aligned primitives, leading to a quadratic growth in primitive count as resolution increases. This fundamentally limits their scalability, making high-resolution synthesis such as 4K intractable. We introduce LGTM (**L**ess **G**aussians, **T**exture **M**ore), a feed-forward framework that overcomes this resolution scaling barrier. By predicting compact Gaussian primitives coupled with per-primitive textures, LGTM decouples geometric complexity from rendering resolution. This approach enables high-fidelity 4K novel view synthesis without per-scene optimization, a capability previously out of reach for feed-forward methods, all while using significantly fewer Gaussian primitives. Project page: https://yxlao.github.io/lgtm/.

## 1 INTRODUCTION

Reconstructing complex scenes and rendering high-fidelity novel views is a key challenge in computer vision and graphics. Systems addressing this challenge should deliver both efficient *feed-forward reconstruction* capabilities, allowing the model to instantly reconstruct new scenes without

---

[†]Work done during an internship at Apple.
[‡]Project lead.

requiring additional per-scene optimization, and *high-resolution rendering* to capture fine details and ensure visual fidelity. These capabilities are crucial for demanding real-world applications, such as augmented and virtual reality, which require both efficient performance and high visual quality to ensure immersive user experiences.

High-resolution feed-forward reconstruction remains challenging. Existing feed-forward 3DGS methods Charatan et al. (2024); Chen et al. (2024a); Fan et al. (2024); Smart et al. (2024); Ye et al. (2025) operate at resolutions in the hundreds. As Gaussian counts grow quadratically with image size (e.g., scaling from 512 to 4K requires 64× more Gaussians), network prediction and Gaussian rendering become prohibitively expensive at high resolutions. Additionally, standard 3DGS couples appearance and geometry within each primitive, requiring an excessive number of Gaussians to represent rich texture regions even on geometrically simple surfaces. While textured Gaussian methods Xu et al. (2024c); Chao et al. (2025); Rong et al. (2024); Song et al. (2024); Weiss & Bradley (2024); Svitov et al. (2025); Xu et al. (2024b) have been proposed to reduce primitive counts, they still require per-scene optimization and cannot generalize across scenes in a feed-forward manner.

To address these challenges, we introduce LGTM, a feed-forward network that predicts textured Gaussians for high-resolution novel view synthesis. Our key idea is to decouple the predictions of geometry parameters and per-primitive textures using a dual-network architecture. LGTM addresses the resolution scalability issue in prior 3DGS feed-forward methods, as well as the per-scene optimization requirement of existing textured Gaussian techniques. Within the dual-network architecture, a primitive network processes low-resolution inputs to predict a compact set of geometric primitives, while a texture network processes high-resolution inputs to predict detailed per-primitive texture maps. The texture network extracts high-resolution features via image patchification and projective mapping, then fuses them with geometric features from the primitive network. We adopt a staged training strategy: we first pre-train the primitive network to establish a robust geometric foundation, and then jointly train it with the texture network to enrich the appearance with high-frequency details. Our framework is also versatile, operating with or without known camera poses. In summary, our contributions are as follows:

- LGTM is the first feed-forward network that predicts textured Gaussians.
- LGTM decouples geometry and appearance through a dual-network architecture. By predicting a compact set of geometric primitives and rich per-primitive textures, it achieves high-resolution rendering (up to 4K) with significantly fewer primitives than prior feed-forward methods.
- LGTM is broadly applicable and can be used with various baseline methods. We demonstrate its improvement on monocular, two-view and multi-view methods, with or without camera poses.

## 2 RELATED WORK

**Feed-forward 3D reconstruction.** Neural Radiance Fields (NeRF) Mildenhall et al. (2020) represents an important advancement in novel view synthesis with neural representations, but its per-scene optimization limits practicality. To address this, generalizable methods Yu et al. (2021); Wang et al. (2021); Chen et al. (2021); Johari et al. (2022) learn cross-scene priors for faster inference. 3D Gaussian Splatting (3DGS) Kerbl et al. (2023) enables real-time rendering via explicit primitives but still requires per-scene training, motivating generalizable 3DGS variants Zou et al. (2024); Charatan et al. (2024); Chen et al. (2024a); Wewer et al. (2024); Chen et al. (2024b); Xu et al. (2024a); Zhang et al. (2024) that directly predict Gaussian parameters from posed images. To remove pose dependency, recent works Wang et al. (2024); Leroy et al. (2024) jointly infer poses and point maps, inspiring pose-free Gaussian splatting Fan et al. (2024); Smart et al. (2024); Ye et al. (2025) and methods that handle more views Wang & Agapito (2025); Tang et al. (2025); Wang et al. (2025b); Zhang et al. (2025); Wang et al. (2025a). However, these feed-forward methods predict pixel-aligned point maps or Gaussians at resolutions in the hundreds. While images from modern cameras are typically 4K or higher, naively scaling up network resolution results in substantial computational and memory costs, limiting real-world applications.

**Textured Gaussian splatting.** Traditional 3DGS Kerbl et al. (2023) and 2DGS Huang et al. (2024) achieve high-quality view synthesis by optimizing Gaussian primitives, but their coupling of appearance and geometry is limiting. Since each Gaussian encodes only a single (view-dependent) color, representing high-frequency textures or complex reflectance demands an excessive number

| Rendering Resolution | Method | Primitive Resolution | Texture Size | Metrics | | | Train Memory (bs = 1) |
|---|---|---|---|---|---|---|---|
| | | | | LPIPS↓ | SSIM↑ | PSNR↑ | |
| 1024×576 | NoPoSplat | 1024×576 | – | 0.239 | 0.716 | 23.169 | 61.85 GB |
| | LGTM | 512×288 | 2×2 | 0.213 | 0.816 | 25.606 | 20.16 GB |
| | LGTM | 256×144 | 4×4 | 0.283 | 0.762 | 24.135 | 16.26 GB |
| 2048×1152 | NoPoSplat | 2048×1152 | – | ✗ | ✗ | ✗ | OOM |
| | LGTM | 512×288 | 4×4 | 0.176 | 0.810 | 25.328 | 21.39 GB |
| | LGTM | 256×144 | 8×8 | 0.246 | 0.759 | 23.628 | 17.46 GB |
| 4096×2304 | NoPoSplat | 4096×2304 | – | ✗ | ✗ | ✗ | OOM |
| | LGTM | 512×288 | 8×8 | 0.200 | 0.803 | 24.489 | 28.23 GB |

Table 1: **LGTM enables high-resolution feed-forward Gaussian splatting.** We compare LGTM with NoPoSplat Ye et al. (2025) variants using different primitive resolutions but yielding the same effective output resolution. NoPoSplat fails to train at 2K or higher due to memory and compute limits, whereas LGTM reaches equivalent resolutions with compact geometric primitives and per-primitive texture maps. Training memory reports peak GPU usage with batch size 1, 2 context views, and 4 target views. For inference performance and memory analysis, see Table 4.

of Gaussians, even for simple geometry (e.g., a flat textured surface). To improve the efficiency of appearance representation, recent works explore integrating texture representations. One strategy involves using global UV texture atlases shared by all Gaussian primitives Xu et al. (2024c). However, optimizing such global texture maps can be challenging for scenes with complex geometric topologies. A more flexible approach employs per-primitive texturing by assigning individual textures to each Gaussian. This includes 3DGS-based approaches Chao et al. (2025); Held et al. (2025) and 2DGS-based ones Rong et al. (2024); Song et al. (2024); Weiss & Bradley (2024); Svitov et al. (2025); Xu et al. (2024b). The type of texture information used in these per-primitive methods varies. Some methods employ standard RGB textures Rong et al. (2024); Song et al. (2024); Weiss & Bradley (2024), some introduce additional opacity maps Chao et al. (2025); Svitov et al. (2025), while others utilize spatially-varying functions for color and opacity instead Xu et al. (2024b); Held et al. (2025). While these textured approaches effectively decouple appearance and geometry for high-fidelity rendering, they require per-scene optimization, meaning a separate optimization process must be performed for each new scene.

## 3 PILOT STUDY

We conducted a pilot study to validate the motivation behind our work: addressing the resolution scalability bottleneck of feed-forward methods. As shown in Table 1, when scaling NoPoSplat Ye et al. (2025) to output 1024×576 primitives, training memory already reaches 61.85 GB for a batch size of 1. Moreover, training fails entirely at 2K and 4K resolutions due to memory constraints.

LGTM trains successfully at 2K and 4K using under 30 GB of memory (Table 1) and scales efficiently at inference: a 64× pixel increase adds only modest memory and time overhead (Table 4). This is enabled by decoupling geometry from appearance – LGTM maintains a compact set of geometric primitives and scales per-primitive textures to reach higher resolutions. This approach exploits the natural frequency separation in scenes: low-frequency geometry vs. high-frequency appearance. Moreover, LGTM offers a tunable trade-off between primitive size and texture size.

## 4 METHOD

LGTM provides a general framework that can be applied to multiple baseline methods with different input settings, such as monocular (Flash3D Szymanowicz et al. (2025)), posed two-view (DepthSplat Xu et al. (2024a)), unposed two-view (NoPoSplat Ye et al. (2025)), and multi-view (VGGT Wang et al. (2025a)) inputs. The LGTM feed-forward network $f$ predicts a set of textured 2D Gaussians from a set of input images $\{I^v\}_{v=1}^N$ and its low-resolution counterparts $\{I_{\text{low}}^v\}_{v=1}^N$:

$$f : (I^v, I_{\text{low}}^v) \rightarrow \{\mu_i, s_i, r_i, c_i, T_i^c, T_i^\alpha\}.$$

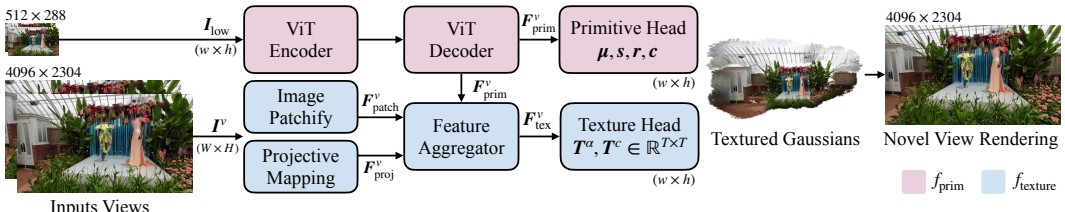

Figure 2: **LGTM Architecture.** *Top*: The primitive network $f_{\text{prim}}$ takes low-resolution images as input and predicts compact geometric primitives $\boldsymbol{\mu}$, $\boldsymbol{s}$, $\boldsymbol{r}$, $\boldsymbol{c}$. *Bottom*: The texture network $f_{\text{texture}}$ processes high-resolution images through image patchify and projective mapping networks, and predicts per-primitive texture maps $\boldsymbol{T}^{\alpha}, \boldsymbol{T}^{c}$. This decoupling of geometry and appearance enables LGTM to achieve feed-forward 4K Gaussian splatting with significantly fewer primitives.

The network is composed of two main submodules: a primitive network, $f_{\text{prim}}$, that predicts compact 2DGS geometric primitives, and a texture network, $f_{\text{texture}}$, that predicts rich texture details. We first introduce the preliminaries of 2DGS and textured Gaussian splatting, and then present the details of our LGTM framework.

## 4.1 PRELIMINARIES

**2D Gaussian Splatting.** 2D Gaussian Splatting (2DGS) Huang et al. (2024) represents scenes with a set of 2D Gaussian primitives in 3D space. To render an image from view $v$, for each pixel coordinate $\boldsymbol{x} = (x, y)$, we find its local coordinates $\boldsymbol{u} = (u, v)$ on a primitive's plane by computing the ray-splat intersection: the intersection point of the camera ray passing through $\boldsymbol{x}$ with primitive $i$. This process is encapsulated by a 2D homography transformation $\boldsymbol{M}$, where $\boldsymbol{u} = \boldsymbol{M}_{i,v}(\boldsymbol{x})$. The local coordinate $\boldsymbol{u}$ is then used to evaluate a 2D Gaussian function $\mathcal{G}(\boldsymbol{u}) = \exp(-\frac{1}{2}(u^2 + v^2))$. The alpha value $a_i$ and color $\hat{c}_i$ for this sample are given by:

$$a_i(\boldsymbol{u}) = o_i \cdot \mathcal{G}(\boldsymbol{u}),$$
$$\hat{\boldsymbol{c}}_i(\boldsymbol{d}_i) = \text{SH}(\boldsymbol{c}_i, \boldsymbol{d}_i), \tag{1}$$

where $\boldsymbol{d}_i$ is the view direction. The final pixel color $\boldsymbol{C}$ is computed by alpha-blending the contributions from all primitives sorted by depth: $\boldsymbol{C} = \sum_i a_i(\boldsymbol{u}_i)\hat{\boldsymbol{c}}_i(\boldsymbol{d}_i) \prod_{j=1}^{i-1}(1 - a_j(\boldsymbol{u}_j))$.

**Textured Gaussian Splatting.** Inspired by classic billboard techniques Décoret et al. (2003) and following the recent BBSplat work Svitov et al. (2025), we augment the standard 2DGS primitive with learnable, per-primitive texture maps: a color texture $\boldsymbol{T}_i^c \in \mathbb{R}^{T \times T \times 3}$ and an alpha texture $\boldsymbol{T}_i^\alpha \in \mathbb{R}^{T \times T}$, where $T$ is the texture resolution. At a ray-splat intersection point $\boldsymbol{u}$, we retrieve color and alpha values from these maps using bilinear sampling (Sec. A.2), denoted by the bracket notation $\boldsymbol{T}[\boldsymbol{u}]$. The alpha texture replaces the Gaussian falloff, and the color texture adds detail to the SH base color. The sample's alpha and color are thus redefined as:

$$a_i(\boldsymbol{u}) = o_i \cdot \boldsymbol{T}_i^\alpha[\boldsymbol{u}],$$
$$\hat{\boldsymbol{c}}_i(\boldsymbol{u}, \boldsymbol{d}_i) = \boldsymbol{T}_i^c[\boldsymbol{u}] + \text{SH}(\boldsymbol{c}_i, \boldsymbol{d}_i). \tag{2}$$

## 4.2 FEED-FORWARD PREDICTION OF TEXTURED GAUSSIANS

LGTM employs a dual-network architecture that decouples geometry and appearance prediction, as illustrated in Fig. 2. The primitive network $f_{\text{prim}}$ takes low-resolution images as input and predicts compact geometric primitives $\boldsymbol{\mu}$, $\boldsymbol{s}$, $\boldsymbol{r}$, $\boldsymbol{c}$. The texture network $f_{\text{texture}}$ processes high-resolution images through image patchify and projective mapping networks, and predicts per-primitive texture maps $\boldsymbol{T}^{\alpha}, \boldsymbol{T}^{c}$. To stabilize the training, we adopt a staged training recipe: first establish a robust geometric foundation, then introduce textural details.

**2DGS pre-training at high resolution.** In the first stage, we train the primitive network $f_{\text{prim}}$ to predict 2DGS parameters:

$$f_{\text{prim}} : \boldsymbol{I}_{\text{low}}^v \rightarrow \{\boldsymbol{F}_{\text{prim}}^v, \boldsymbol{\mu}_i, \boldsymbol{s}_i, \boldsymbol{r}_i, o_i, \boldsymbol{c}_i\}, \tag{3}$$

where $\boldsymbol{F}_{\text{prim}}^v$ are the feature maps from the ViT decoder that will also be shared with $f_{\text{texture}}$. The primitive network takes a low-resolution version of the input, $\boldsymbol{I}_{\text{low}}^v \in \mathbb{R}^{h \times w \times 3}$, and processes it through a ViT encoder-decoder to predict the scene's geometry and low-frequency appearance, producing a grid of $h \times w$ 2DGS primitives.

The key idea is high-resolution supervision: the network takes low-resolution inputs $\boldsymbol{I}_{\text{low}}^v$ and predicts an $h \times w$ primitive grid, but renders and supervises them at full resolution $H \times W$. Although the predicted Gaussians can be rendered at arbitrary resolutions, doing so without high-res supervision may result in areas with holes, as they are not anti-aliased (see supplementary Sec. A.1 for details). The high-resolution supervision forces the network to learn predictive scales $\boldsymbol{s}$ and other parameters appropriately sized for high-resolution rendering, establishing a strong geometric prior.

**Learned projective texturing.** The texture network $f_{\text{texture}}$ takes in high-resolution images as well as low-resolution primitive features $\boldsymbol{F}_{\text{prim}}^v$ and predicts per-primitive texture maps $\boldsymbol{T}^\alpha, \boldsymbol{T}^c$:

$$f_{\text{texture}} : (\boldsymbol{I}^v, \boldsymbol{F}_{\text{prim}}^v) \rightarrow \{\boldsymbol{T}_i^c, \boldsymbol{T}_i^\alpha\}. \tag{4}$$

At a high level, $f_{\text{texture}}$ combines three complementary features: $\boldsymbol{F}_{\text{patch}}^v$ is computed from the patchified high-resolution image followed by convolutional layers to encode local features; the projective features $\boldsymbol{F}_{\text{proj}}^v$ are extracted from projective prior textures, which provide strong high-frequency texture details; and $\boldsymbol{F}_{\text{prim}}^v$ reuses backbone features. These features $(\boldsymbol{F}_{\text{prim}}^v, \boldsymbol{F}_{\text{patch}}^v, \boldsymbol{F}_{\text{proj}}^v)$ are aggregated to predict the final per-primitive textures $\{\boldsymbol{T}_i^c, \boldsymbol{T}_i^\alpha\}$.

To compute projective features $\boldsymbol{F}_{\text{proj}}^v$, we perform projective texture mapping from the image back to the textured Gaussian primitives. For each Gaussian primitive $i$, we compute a projective prior texture $\boldsymbol{T}_i^{c,\text{proj}}$ using the inverse transformation $\boldsymbol{M}_{i,v}^{-1} : \boldsymbol{u} \rightarrow \boldsymbol{x}$ that maps from primitive local coordinates $\boldsymbol{u}$ to source image pixel coordinates $\boldsymbol{x}$, and then sample the RGB color from the high-resolution source image $\boldsymbol{I}^v$ at $\boldsymbol{x}$:

$$\boldsymbol{T}_i^{c,\text{proj}}[\boldsymbol{u}] = \boldsymbol{I}^v[\boldsymbol{x}] = \boldsymbol{I}^v[\boldsymbol{M}_{i,v}^{-1}(\boldsymbol{u})]. \tag{5}$$

Intuitively, the projective prior $\boldsymbol{T}_i^{c,\text{proj}}[\boldsymbol{u}]$ is computed by the inverse process of Gaussian rasterization: instead of rendering Gaussians to an image, we "render" the source image back onto the Gaussian texture maps using the inverse transformation. This projection step is highly efficient as it can typically be done in a few milliseconds for a 4K image. Finally, we extract projective features $\boldsymbol{F}_{\text{proj}}^v$ from $\boldsymbol{T}_i^{c,\text{proj}}[\boldsymbol{u}]$ to provide strong high-frequency appearance features for texture prediction.

**Training recipe.** LGTM employs a progressive two-stage training strategy that gradually introduces texture complexity while maintaining geometric stability. In the first stage, we train the primitive network $f_{\text{prim}}$ in isolation to predict 2DGS parameters using low-resolution inputs with high-resolution supervision. In the second stage, we jointly train both the primitive network $f_{\text{prim}}$ and texture network $f_{\text{texture}}$. To maintain geometric stability, the pre-trained primitive network parameters are trained with a reduced learning rate ($0.1\times$). The color texture $\boldsymbol{T}^c$ is zero-initialized and added to the SH base color (Eq. 2) to provide high-frequency color details. Both stages are supervised with standard photometric losses (MSE + LPIPS).

## 5 EXPERIMENTS

### 5.1 EXPERIMENTAL SETUP

**Baselines.** LGTM can be applied to most existing feed-forward Gaussian splatting methods to enable high-resolution novel view synthesis. We evaluate LGTM across three scenarios with the following baseline methods: single-view with Flash3D Szymanowicz et al. (2025), two-view with both the pose-free NoPoSplat Ye et al. (2025) and the posed DepthSplat Xu et al. (2024a), and multi-view with VGGT Wang et al. (2025a). For each scenario, we compare three variants: 3DGS, 2DGS, and LGTM. For the 3DGS and 2DGS baselines, we re-train them with high-resolution supervision (Sec. A.1). We train and evaluate across multiple resolutions and report standard image quality metrics: LPIPS↓, SSIM↑, and PSNR↑.

**Datasets.** We evaluate LGTM with RealEstate10K (RE10K) Zhou et al. (2018) and DL3DV-10K Ling et al. (2024). For RE10K, we follow the official train-test split consistent with prior

| Dataset | Method | | Render Resolution | Primitive Resolution | Texture Size | Metrics | | |
|---------|--------|--|-------------------|----------------------|--------------|---------|--|--|
| | Baseline | Primitive | | | | LPIPS↓ | SSIM↑ | PSNR↑ |
| RE10K | NoPoSplat | 3DGS | 1024×576 (1K) | 512×288 | – | 0.250 | 0.823 | 24.666 |
| | | 2DGS | 1024×576 (1K) | 512×288 | – | 0.225 | 0.829 | 24.674 |
| | | LGTM | 1024×576 (1K) | 512×288 | 2×2 | **0.195** | **0.847** | **25.419** |
| | NoPoSplat | 3DGS | 2048×1152 (2K) | 512×288 | – | 0.257 | 0.819 | 24.235 |
| | | 2DGS | 2048×1152 (2K) | 512×288 | – | 0.227 | 0.823 | 24.282 |
| | | LGTM | 2048×1152 (2K) | 512×288 | 4×4 | **0.182** | **0.856** | **25.233** |
| DL3DV | NoPoSplat | 3DGS | 1024×576 (1K) | 512×288 | – | 0.297 | 0.747 | 24.131 |
| | | 2DGS | 1024×576 (1K) | 512×288 | – | 0.260 | 0.736 | 23.569 |
| | | LGTM | 1024×576 (1K) | 512×288 | 2×2 | **0.213** | **0.816** | **25.606** |
| | NoPoSplat | 3DGS | 2048×1152 (2K) | 512×288 | – | 0.292 | 0.737 | 23.814 |
| | | 2DGS | 2048×1152 (2K) | 512×288 | – | 0.262 | 0.731 | 23.427 |
| | | LGTM | 2048×1152 (2K) | 512×288 | 4×4 | **0.176** | **0.810** | **25.328** |
| | DepthSplat | 3DGS | 1920×1024 (2K) | 960×512 | – | 0.161 | 0.826 | 25.705 |
| | | 2DGS | 1920×1024 (2K) | 960×512 | – | 0.164 | 0.823 | 25.967 |
| | | LGTM | 1920×1024 (2K) | 960×512 | 2×2 | **0.159** | **0.832** | **26.218** |
| | NoPoSplat | 3DGS | 4096×2304 (4K) | 512×288 | – | 0.351 | 0.753 | 23.022 |
| | | 2DGS | 4096×2304 (4K) | 512×288 | – | 0.322 | 0.737 | 22.198 |
| | | LGTM | 4096×2304 (4K) | 512×288 | 8×8 | **0.200** | **0.803** | **24.489** |
| | DepthSplat | 3DGS | 3840×2048 (4K) | 960×512 | – | 0.210 | 0.801 | 24.740 |
| | | 2DGS | 3840×2048 (4K) | 960×512 | – | 0.198 | 0.794 | 24.715 |
| | | LGTM | 3840×2048 (4K) | 960×512 | 4×4 | **0.170** | **0.827** | **25.508** |

Table 2: **Novel view synthesis results under two-view setting.** Performance of LGTM over NoPoSplat and DepthSplat across rendering resolutions and datasets. For 4K resolution, we use 4096×2304 for NoPoSplat and 3840×2048 for DepthSplat due to the requirements of their backbone networks.

work Charatan et al. (2024), and report results up to 2K resolution[1]. For DL3DV, we use the benchmark subset for testing and the remaining data for training, and report results up to 4K resolution to demonstrate high-resolution feed-forward reconstruction and novel view synthesis.

## 5.2 Main Results

**Two-view.** Table 2 presents our main results for two-view novel view synthesis on the RE10K and DL3DV datasets. We evaluate LGTM with two baselines: pose-free NoPoSplat and posed DepthSplat. For both baselines, LGTM consistently outperforms the 3DGS and 2DGS variants across all tested resolutions and all metrics. A similar trend is observed on the higher-resolution DL3DV dataset, where LGTM again consistently surpasses baseline performance at 4K. Beyond improvements on pixel-wise metrics PSNR and SSIM, LGTM shows stronger improvement on the perceptual metric LPIPS with a 23% – 75% reduction. Fig. 3 shows novel views synthesized by LGTM and the baseline methods on the DL3DV dataset at 4K resolution. The comparison between baselines and LGTM indicates that LGTM is general and effective for modeling high-frequency details with compact texture maps, leading to higher-fidelity renderings.

**Single-view.** Table 3 shows results for single-view novel view synthesis on the DL3DV dataset. With Flash3D Szymanowicz et al. (2025) as the baseline, LGTM again achieves the best performance on all metrics at all resolutions. Fig. 4 provides a qualitative comparison showing that LGTM renders finer details and textures, which are often blurred or lost by baseline methods due to their limited number of geometric primitives. Notably, LGTM achieves high-quality renderings with only 512×288 geometric primitives, strongly demonstrating the power of rich per-primitive textures.

---

[1]The terms "2K" and "4K" refer to horizontal resolutions of approximately 2,000 and 4,000 pixels, respectively. RE10K offers 2K 1920×1080 resolution (also commonly known as 1080p) for a subset of its video sources, while DL3DV-10K offers 4K resolutions at 3840×2160.

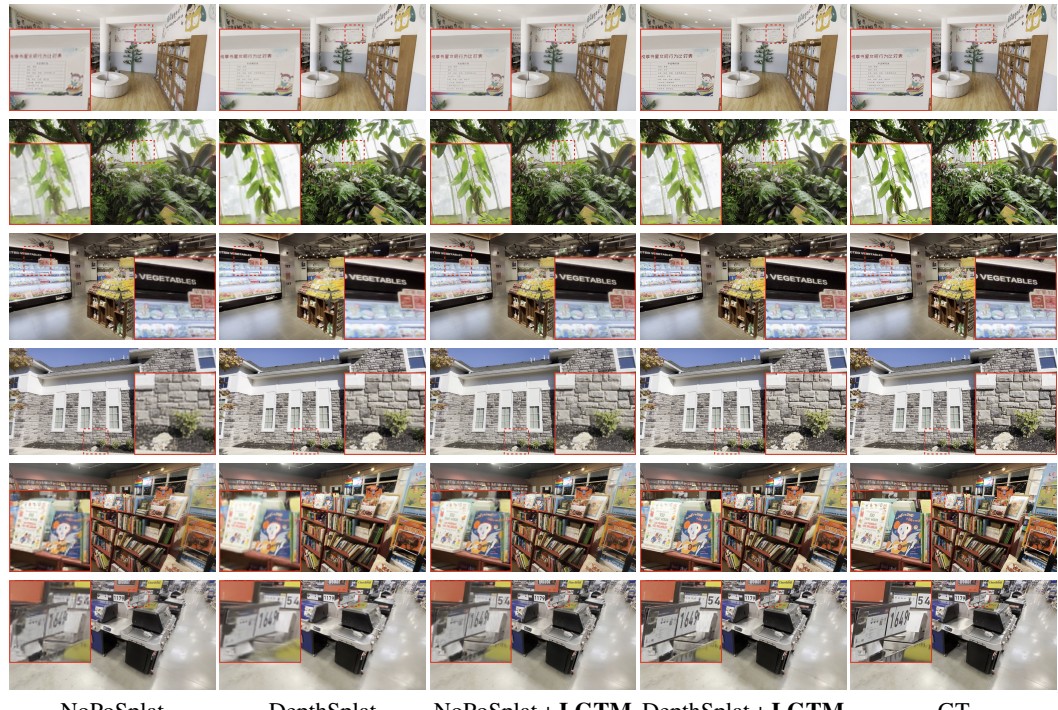

NoPoSplat          DepthSplat     NoPoSplat + **LGTM**  DepthSplat + **LGTM**          GT

Figure 3: **Qualitative comparison in the two-view setting.** Evaluated on the DL3DV dataset at 4K resolution. Best viewed in color and zoomed in.

| Inputs | Method | | Render Resolution | Primitive Resolution | Texture Size | Metrics | | |
|---|---|---|---|---|---|---|---|---|
| | Baseline | Primitive | | | | LPIPS↓ | SSIM↑ | PSNR↑ |
| Single-view | Flash3D | 3DGS | 1024×576 (1K) | 512×288 | – | 0.274 | 0.672 | 21.353 |
| | | 2DGS | 1024×576 (1K) | 512×288 | – | 0.267 | 0.676 | 21.447 |
| | | LGTM | 1024×576 (1K) | 512×288 | 2×2 | **0.215** | **0.708** | **21.910** |
| | Flash3D | 3DGS | 2048×1152 (2K) | 512×288 | – | 0.368 | 0.643 | 19.911 |
| | | 2DGS | 2048×1152 (2K) | 512×288 | – | 0.358 | 0.653 | 20.276 |
| | | LGTM | 2048×1152 (2K) | 512×288 | 4×4 | **0.224** | **0.712** | **21.552** |
| | Flash3D | 3DGS | 4096×2304 (4K) | 512×288 | – | 0.399 | 0.724 | 20.068 |
| | | 2DGS | 4096×2304 (4K) | 512×288 | – | 0.371 | 0.725 | 20.322 |
| | | LGTM | 4096×2304 (4K) | 512×288 | 8×8 | **0.219** | **0.766** | **21.778** |
| Multi-view | VGGT | 3DGS | 1036×560 (1K) | 518×280 | – | 0.336 | 0.649 | 20.278 |
| | | 2DGS | 1036×560 (1K) | 518×280 | – | 0.341 | 0.646 | 20.177 |
| | | LGTM | 1036×560 (1K) | 518×280 | 2×2 | **0.325** | **0.676** | **20.645** |
| | VGGT | 3DGS | 2072×1120 (2K) | 518×280 | – | 0.380 | 0.644 | 19.461 |
| | | 2DGS | 2072×1120 (2K) | 518×280 | – | 0.392 | 0.641 | 19.273 |
| | | LGTM | 2072×1120 (2K) | 518×280 | 4×4 | **0.361** | **0.661** | **19.990** |

Table 3: **Single-view and multi-view novel view synthesis results.** Comparison of LGTM with Flash3D Szymanowicz et al. (2025) and VGGT Wang et al. (2025a) baselines across different view settings and resolutions on DL3DV dataset.

**Multi-view.** To demonstrate that LGTM is a general framework supporting different numbers of views as input, we build a 4-view feed-forward LGTM variant with VGGT Wang et al. (2025a). We implement a Gaussian prediction head over the VGGT backbone following AnySplat Jiang et al. (2025) to serve as our baseline. During training, we align predicted camera poses with ground truth poses to mitigate potential misalignment between rendered novel views and ground truth. As shown in Table 3, LGTM achieves consistent improvements at 1K and 2K resolutions. We did not scale up

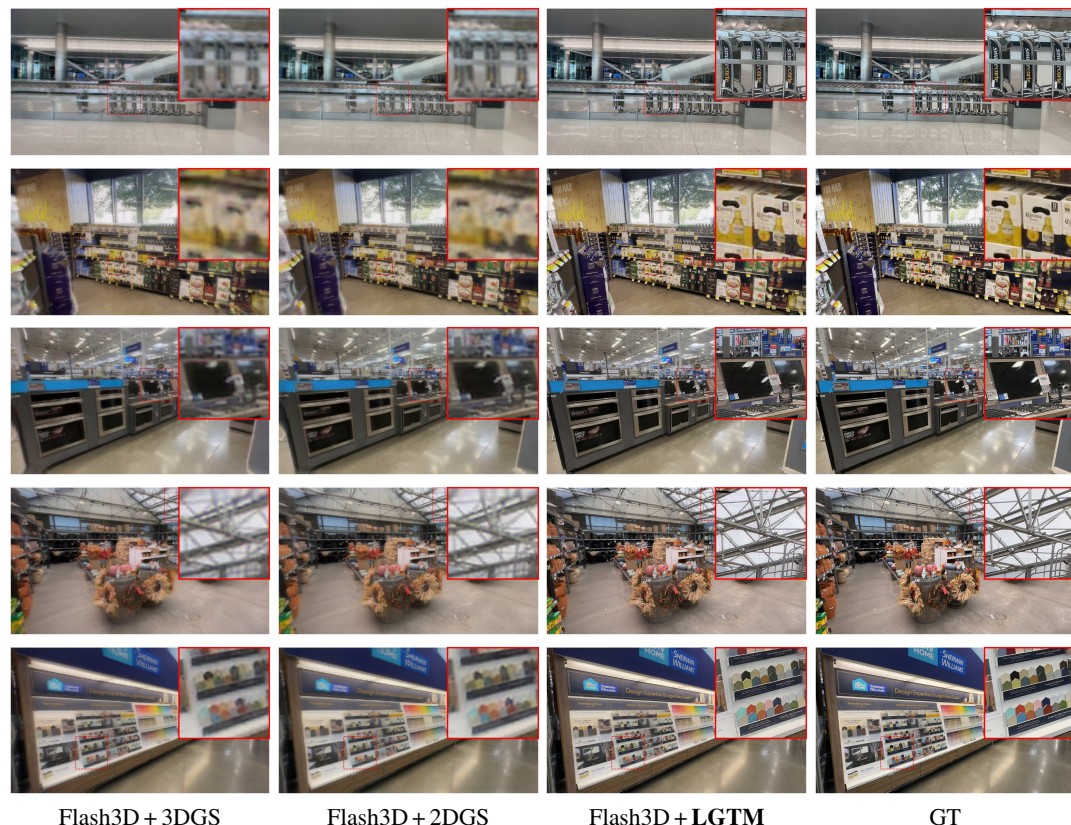

|                    |                    |                         |      |
|--------------------|--------------------|-------------------------|------|
| Flash3D + 3DGS     | Flash3D + 2DGS     | Flash3D + **LGTM**      | GT   |

Figure 4: **Qualitative comparison in the single-view setting.** We re-train Flash3D on DL3DV with the same number of geometric primitives (512×288) for 3DGS, 2DGS, and LGTM.

to 4K resolution due to memory constraints even with the VGGT backbone frozen, which we leave for future work.

**Performance benchmark.** Table 4 presents a detailed inference performance analysis for the two-view synthesis task, where two views are input to predict a single target view. We benchmark on a single NVIDIA A100 GPU using a batch size of one; for each case, we report peak memory and average timings over 10 runs after 3 warmups. LGTM is highly efficient when scaling to high resolutions. This is highlighted by comparing the NoPoSplat 512×288 2DGS model (②) with the LGTM 4096×2304 model (⑤): for a 64× increase in pixels, LGTM requires only 1.80× the peak memory and 1.47× the total time. The scalability advantage of LGTM's texture-based upsampling becomes increasingly pronounced at higher resolutions, where traditional methods face prohibitive costs. While Table 4 shows inference performance, Table 1 analyzes training memory requirements.

|     | Method         | Primitive Resolution | Texture Size | Render Resolution | Peak GPU Memory (GB) | Network Fwd. Time (ms) | Render Time (ms) | Total Time (ms) |
|-----|----------------|----------------------|--------------|-------------------|----------------------|------------------------|------------------|-----------------|
| ① | NoPoSplat 3DGS | 512×288              | –            | 512×288           | 3.06                 | 110.37                 | 3.81             | 114.18          |
| ② | NoPoSplat 2DGS | 512×288              | –            | 512×288           | 3.06                 | 112.70                 | 6.43             | 119.13          |
| ③ | LGTM           | 512×288              | 2×2          | 1024×576          | 4.33                 | 132.06                 | 7.70             | 139.76          |
| ④ | LGTM           | 512×288              | 4×4          | 2048×1152         | 4.60                 | 136.14                 | 14.26            | 150.40          |
| ⑤ | LGTM           | 512×288              | 8×8          | 4096×2304         | 5.51                 | 142.28                 | 32.82            | 175.10          |

Table 4: **Inference time and memory benchmark.** LGTM is highly efficient when scaling up to high resolutions: compared to the NoPoSplat 512×288 2DGS (②), the LGTM 4096×2304 model (⑤) represents a 64× increase in pixels, yet requires only 1.80× the peak memory and 1.47× the total time. For an analysis of training memory requirements, see Table 1.

| | Method | LPIPS↓ | SSIM↑ | PSNR↑ |
|---|---|---|---|---|
| ① | NoPoSplat (3DGS) | 0.371 | 0.583 | 16.964 |
| ② | + High-res retrain (2DGS) | 0.256 | 0.731 | 23.502 |
| ③ | + Image patchify only | 0.199 | 0.782 | 24.673 |
| ④ | + Texture color | 0.189 | 0.806 | 25.314 |
| ⑤ | + Texture alpha (full model) | 0.176 | 0.810 | 25.328 |

Table 5: **Ablation study on LGTM components.** We start from the NoPoSplat baseline and progressively integrate components of LGTM, evaluated on DL3DV at 2K resolution. All models operate on 512×288 primitives. The baseline is trained on low-resolution and evaluated at 2K. Subsequent versions are trained with high-resolution supervision.

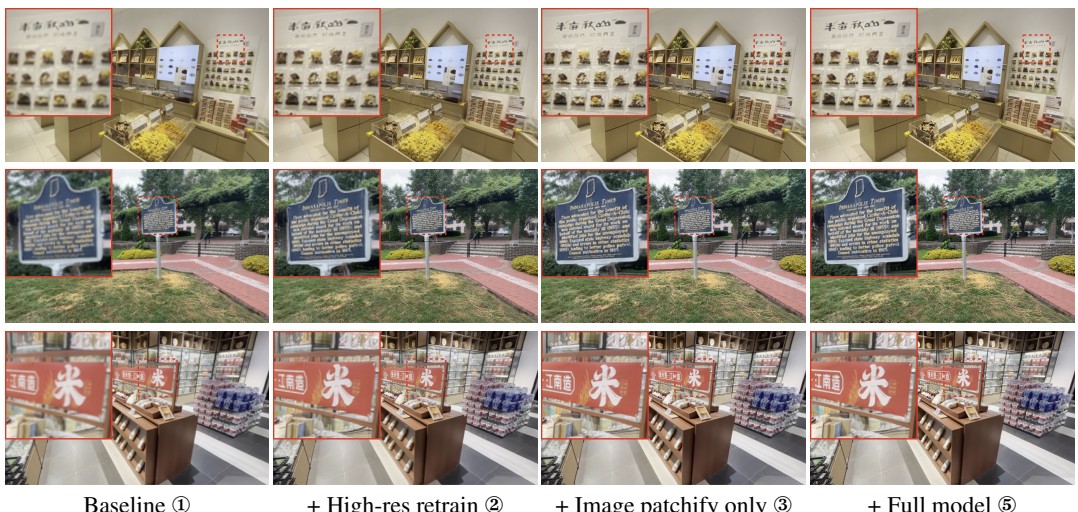

Baseline ①      + High-res retrain ②      + Image patchify only ③      + Full model ⑤

Figure 5: **Qualitative results of the ablation study.** Each image shows the progressive improvement as components of LGTM are added to the NoPoSplat baseline. The full model produces renderings that are visibly closer to the ground truth.

## 5.3 ABLATION STUDY

We conduct an ablation study to analyze the contribution of each component of LGTM, with results presented in Table 5. Our base model (①) is NoPoSplat Ye et al. (2025) using 3D Gaussians, trained on low-resolution inputs and evaluated at 2K, which yields poor performance. Simply applying high-resolution supervision (②) significantly improves results, establishing a stronger baseline for comparison. From this improved baseline, we introduce the core components of LGTM. First, adding image patchified features (③) provides a substantial boost across all metrics, demonstrating effectiveness in capturing high-frequency details. We then add a learned texture color map (④), which further improves performance by enriching the appearance details. Finally, incorporating a learned texture alpha map in the full LGTM model (⑤) yields the best results, confirming that both texture color and alpha are essential for high-quality rendering. The qualitative comparison in Fig. 5 visually reinforces these findings, showing clear progression in rendering quality.

## 6 CONCLUSION

We introduce LGTM, a feed-forward network that predicts textured Gaussians for high-resolution rendering. LGTM addresses the resolution scalability barrier that has limited feed-forward 3DGS methods to low resolutions. LGTM achieves 4K novel view synthesis where traditional approaches fail due to memory constraints, requiring only 1.80× memory and 1.47× time for a 64× increase in pixel count. The consistent improvements across multiple baseline methods (Flash3D, NoPoSplat, DepthSplat, VGGT) demonstrate the broad applicability of our approach.

**Limitations.** While LGTM addresses texture scaling well, reconstruction quality still depends heavily on geometry. Empirically, LGTM performs best in the single-view setting (Flash3D) without multi-view inconsistency, better in the posed two-view setting (DepthSplat) than the unposed one (NoPoSplat) due to improved geometry, and shows marginal gains in the multi-view setting (VGGT) where geometry is less precise. Additionally, our current framework operates with pre-defined texture resolutions, requiring manual tuning of texture size to balance quality and computational cost.

**Acknowledgments.** This work is supported by the Hong Kong Research Grant Council General Research Fund (No. 17213925) and National Natural Science Foundation of China (No. 62422606).

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

# LESS GAUSSIANS, TEXTURE MORE:
# 4K FEED-FORWARD TEXTURED SPLATTING

## SUPPLEMENTARY MATERIALS

## TABLE OF CONTENTS

## A.1 HIGH-RESOLUTION RE-TRAINING

Baseline methods such as NoPoSplat and DepthSplat are typically trained at low resolution (up to 1K). When building a high-resolution LGTM variant from the baseline, we first re-train the baseline with high-resolution supervision. This is crucial for the network to learn primitive scales and other attributes properly for high-resolution rendering, serving as a strong geometry prior for texture learning. Here we demonstrate the effectiveness of high-resolution re-training by comparing three methods: *Direct Render* directly renders the Gaussians produced by low-resolution baseline models at the target resolution; *Render and Upsample* renders the output Gaussians at training resolution, then upsamples the resulting image to the target resolution; *High-Resolution Re-train* re-trains the baseline model by rendering and supervising at the target resolution.

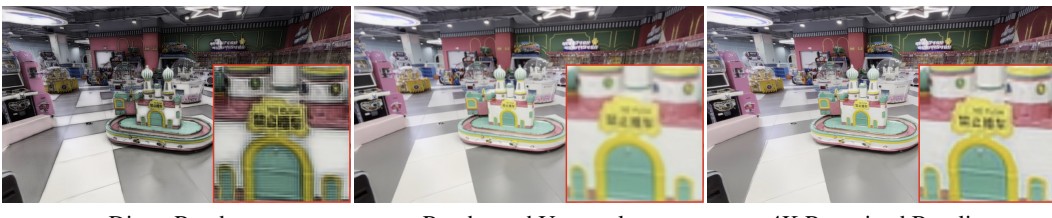

Direct Render      Render and Upsample      4K Re-trained Baseline

Figure 6: **Effects of high-resolution re-training.**

As shown in Fig. 6, direct rendering of Gaussians at higher resolution produces holes because 3DGS lacks anti-aliasing. Rendering and upsampling eliminates holes but produces blurry results since the effective image resolution remains low. *Re-train* produces the best results among the three methods, demonstrating the effectiveness and necessity of high-resolution re-training. This strong geometry prior then serves as the foundation for LGTM to learn textural details. *Re-train* produces slightly better results than *Render and Upsample* but still suffers from blurriness due to the limited texture complexity of vanilla 3DGS. In Table 2, we use the render and upsample strategy to evaluate baseline methods.

## A.2 BILINEAR TEXTURE SAMPLING

Given a texture map $T$ of resolution $T \times T$ and a ray-splat intersection point with local coordinates $(u, v)$ on the primitive plane, we retrieve the texture value via bilinear sampling with border clamping, as detailed in Algorithm 1. The scale $\sigma$ is a shared hyperparameter that controls the span of the texture region within the 2DGS primitive.

---

**Algorithm 1** Bilinear texture sampling at local coordinates $(u, v)$

---

1: **function** BILINEAR-SAMPLE$(\boldsymbol{T}, T, u, v, \sigma)$
2:     $u \leftarrow \frac{(u+\sigma)}{2\sigma} \cdot T - 0.5$          ▷ Transform to pixel coordinates
3:     $v \leftarrow \frac{(v+\sigma)}{2\sigma} \cdot T - 0.5$
4:     $u \leftarrow \mathrm{clamp}(u, 0, T-1)$          ▷ Border clamping
5:     $v \leftarrow \mathrm{clamp}(v, 0, T-1)$
6:     $i_u \leftarrow \lfloor u \rfloor, \quad f_u \leftarrow u - i_u$          ▷ Integer and fractional parts
7:     $i_v \leftarrow \lfloor v \rfloor, \quad f_v \leftarrow v - i_v$
8:     $i_{u+1} \leftarrow \min(i_u + 1, T-1), \quad i_{v+1} \leftarrow \min(i_v + 1, T-1)$          ▷ Corner indices
9:     **return** $(1 - f_u)(1 - f_v) \cdot \boldsymbol{T}[i_u, i_v] + f_u(1 - f_v) \cdot \boldsymbol{T}[i_{u+1}, i_v]$
10:         $+ (1 - f_u)f_v \cdot \boldsymbol{T}[i_u, i_{v+1}] + f_u f_v \cdot \boldsymbol{T}[i_{u+1}, i_{v+1}]$
11: **end function**

---

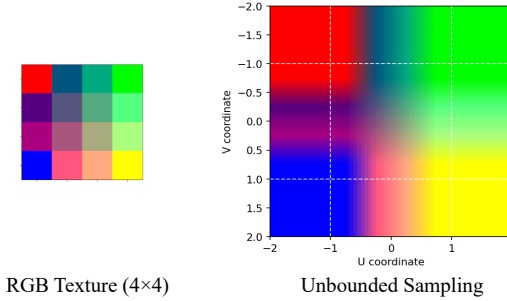

RGB Texture (4×4)          Unbounded Sampling

Figure 7: **Unbounded bilinear texture sampling.** Our sampling method uses border clamping to handle out-of-bounds coordinates, allowing texture information to extend beyond the primitive boundary. This provides smoother transitions and better coverage compared to bounded sampling approaches that use zero-padding for out-of-bounds areas.

For texture colors, we use an unbounded bilinear sampling strategy that differs from BBSplat Svitov et al. (2025). In BBSplat, color sampling is bounded and the color decays from the Gaussian center, while traditional non-textured 2DGS maintains constant colors with only opacity decay. Our unbounded sampling ensures equal weights across the four bilinear corners, making it closer to the original non-textured 2DGS formulation and allowing us to learn texture colors using only 2DGS opacity as alpha fall-off. This prevents dark edge artifacts that would result from combining BBSplat's bounded sampling with regular 2DGS opacity. Our method is similar to Gaussian Billboards Weiss & Bradley (2024), where $\sigma$ controls the relative span of the textured region.

## A.3 ROBUSTNESS TO LARGER INPUT VIEW GAPS

To evaluate whether LGTM's textured Gaussian representation works well beyond small viewpoint differences, we assess novel view synthesis quality under different camera pose settings with gradually increasing context view differences. We vary the context view gap on the DL3DV test set to examine how performance changes as the baseline between input views increases.

For each context frame index gap $\{10, 20, 30, 40\}$, we select 2 source views with the specified gap and sample 4 target views evenly distributed within the range for numerical evaluation. Both No-PoSplat baseline and NoPoSplat + LGTM use 512×288 Gaussians per-view with 2 context views, trained and evaluated at 4096×2304 resolution, with the baseline trained with high-resolution supervision (Sec. A.1).

Fig. 8 provides qualitative comparisons across different frame gaps. Each row shows the two context views along with target view renderings from both NoPoSplat baseline and NoPoSplat + LGTM. As the gap increases from 10 to 40 frames, the viewpoint differences become more substantial, making the reconstruction task increasingly challenging. Despite this, NoPoSplat + LGTM consistently produces sharper details and fewer artifacts compared to the baseline across all settings, confirming

the quantitative results. Table 6 shows that NoPoSplat + LGTM consistently outperforms the baseline across all context view gaps. While both methods experience performance degradation as the gap increases, NoPoSplat + LGTM maintains its superiority even at gap 40, where the viewpoint difference is substantial. These results demonstrate that per-primitive texture maps handle novel viewpoints beyond small differences, addressing concerns that the method might only work well for closely spaced input views.

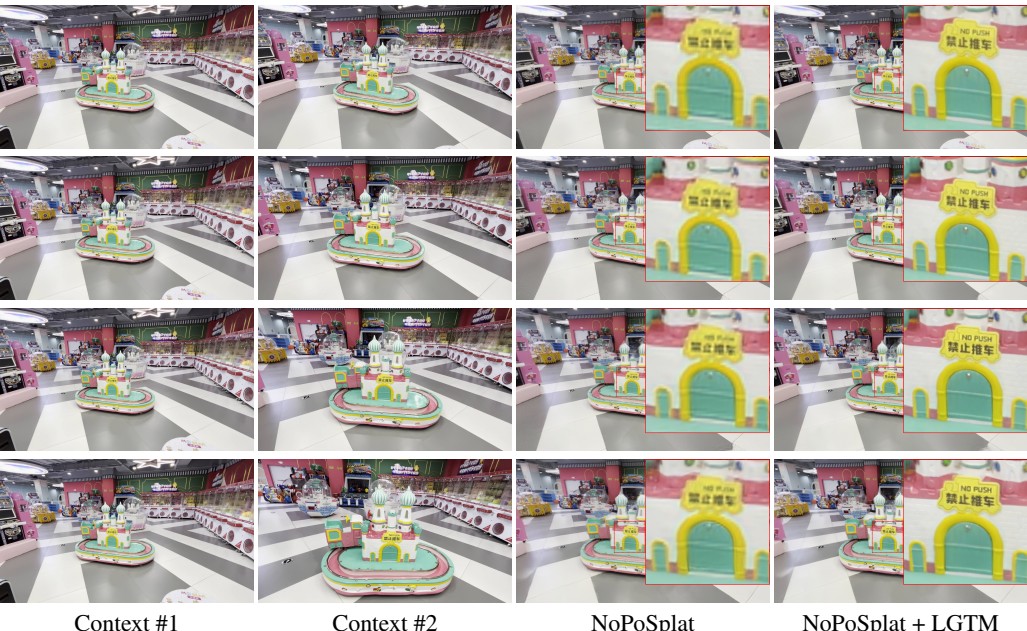

| Context #1 | Context #2 | NoPoSplat | NoPoSplat + LGTM |

Figure 8: **Qualitative comparison across different input view gaps.** Each row shows context frame gaps of 10, 20, 30, and 40 frames. LGTM consistently produces sharper details and fewer artifacts.

| Context view gap | Method | LPIPS↓ | SSIM↑ | PSNR↑ |
|---|---|---|---|---|
| 10 | NoPoSplat | 0.322 | 0.737 | 22.198 |
| | NoPoSplat + LGTM | **0.200** | **0.803** | **24.489** |
| 20 | NoPoSplat | 0.366 | 0.722 | 20.732 |
| | NoPoSplat + LGTM | **0.258** | **0.765** | **22.159** |
| 30 | NoPoSplat | 0.413 | 0.709 | 19.430 |
| | NoPoSplat + LGTM | **0.323** | **0.730** | **20.123** |
| 40 | NoPoSplat | 0.458 | 0.696 | 18.264 |
| | NoPoSplat + LGTM | **0.376** | **0.707** | **18.862** |

Table 6: **Quantitative comparison across different input view gaps.** LGTM consistently outperforms the baseline across all gaps (10, 20, 30, 40 frames), demonstrating robustness to larger viewpoint differences.

## A.4 COMPARISON WITH PER-SCENE OPTIMIZATION

We compare DepthSplat + LGTM with per-scene optimized 3D Gaussian Splatting on DL3DV at 4K. Both methods use 2 context views (frames 0 and 20) and are evaluated on 19 target views (frames 1-19). For per-scene optimization, we run COLMAP structure-from-motion on all frames $\{0, 1, 2, \ldots, 20\}$, as running COLMAP on only 2 context frames fails. We then filter the sparse point cloud to retain only points visible from the context views for initialization. We optimize using the 2 context views at 4K following the default 3DGS protocol, with 30,000 iterations, refinement every 100 iterations, and densification stopping at 15,000 iterations.

Table 7 shows that DepthSplat + LGTM outperforms per-scene optimization across all metrics while being orders of magnitude faster. The optimization-based approach overfits to the context views, as shown in Fig. 9, where its PSNR drops to ~20-22 dB for middle frames while DepthSplat + LGTM maintains stable performance. By training on diverse scenes, DepthSplat + LGTM develops strong priors for generalizing to novel viewpoints, whereas per-scene optimization can only interpolate between limited training views. DepthSplat + LGTM achieves instant reconstruction compared to ~30 minutes for per-scene optimization on a single A100 GPU.

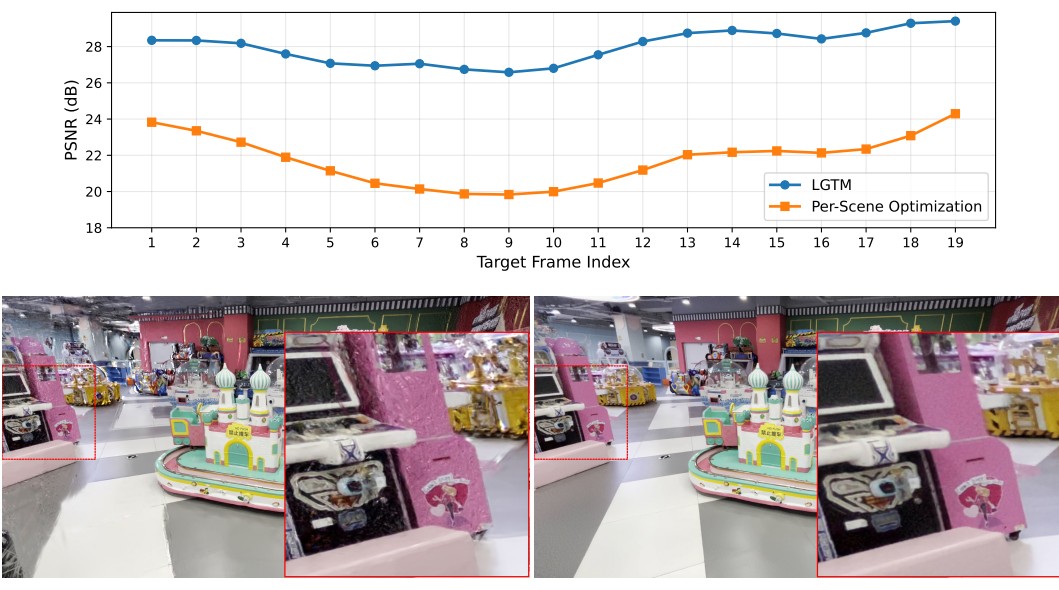

Figure 9: **Comparing LGTM to per-scene optimization.** *Top:* Per-scene optimization overfits to context views with degraded performance on middle frames. DepthSplat + LGTM maintains stable quality across all frames. *Bottom:* Target frame #10. While both achieve similar sharpness in the center, LGTM produces sharper details with fewer artifacts toward the edges.

| Method | Per-Scene Optimization | DepthSplat + LGTM (Ours) |
|---|---|---|
| Frames for COLMAP | $\{0, 1, 2, \ldots, 20\}$ | – |
| Context views (optimization) | $\{0, 20\}$ | $\{0, 20\}$ |
| Target views (evaluation) | $\{1, 2, 3, \ldots, 19\}$ | $\{1, 2, 3, \ldots, 19\}$ |
| PSNR↑ | 21.75 | **27.99** |
| SSIM↑ | 0.78 | **0.88** |
| LPIPS↓ | 0.21 | **0.15** |

Table 7: **Comparison with per-scene optimization.** DepthSplat + LGTM outperforms per-scene optimized 3DGS while being orders of magnitude faster.

## A.5 USE OF LARGE LANGUAGE MODELS

Declaration of use of Large Language Models (LLMs): We use LLMs to help us polish sentences in the manuscript and fix grammar and spelling mistakes.

