# OpenReview forum: "Less Gaussians, Texture More: 4K Feed-Forward Textured Splatting"
_ICLR.cc/2026/Conference — ICLR 2026 Poster_

### Official Review · Reviewer_T5YU · 2025-10-23

**Soundness:** 2
**Presentation:** 2
**Contribution:** 2
**Rating:** 2
**Confidence:** 5

**Summary:**

The paper introduces LGTM (Less Gaussians, Texture More), a feed-forward framework for high-resolution (up to 4K) novel view synthesis with textured Gaussian primitives. The core idea is to decouple geometry from appearance: a primitive network consumes low-resolution inputs to predict a compact grid of 2DGS geometry (centers, scales, rotations, SH base color), while a texture network consumes high-resolution inputs to predict per-primitive color and alpha texture maps via image patchifying and learned projective texturing. They used a 2-stage training, achieving coarse-to-fine prediction. Firstly a rough geometric and then detailed textures.

**Strengths:**

1. The paper is well-writen and easy to understand.
2. The results seem to be pretty good improvement on the base models.
3. Pretty efficient algorithm and is friendly to small GPUs.

**Weaknesses:**

1. This work uses the few Gaussians from low-resolution images to serve as geometry probes, and paint the surfaces with higher resolution images. This is some what equivalent to first obtain all Gaussian points from the full resolution images (one may achieve this through similar way as e.g. Point3R[1]) and then uniformly **drop** most of the Gaussians. This relies on an important (but not necessarily true) assumption that real-world geometries are all smooth. This may causes the model to potentially fail when observing high frequency geometry details e.g. hairs, cloths, bumps, etc. No result has been given in that regard. Instead, a more clever way of downsampling / compressing Gaussians could be leveraged to both preserving geometry details & acceptable resources needed.
2. The baseline models are never trained on high resolution images while it can be done: e.g. augmenting the original data with different zoom-in / zoom-out scale & crop to original resolution at train time and inference with full-resolution. The current large scale evaluation doesn't make too much sense.
3. In L232-245 the authors use literally 1/4 page talking about a very elementary technique on assigning texture colors to Gaussian points, while ignoring some of the possibly more challenging topics such as how to handle view inconsistencies with less Gaussian points on high resolution images.
4. The authors may have used the term "primitive" with different meanings without firmly defining any of them. As a result, it is hard to parse some parts of the manuscript. For example, what is "per-primitive texture maps", and how does it differ from "2DGS primitives"? The term "Primitive Resolution" repeatedly appears but is never explained.
5. The use of mathematical symbols are not consistent, e.g. Eq 4. $f_{texture}$ seems like a mapping, while in L226, it becomes a "feature". The use of superscripts and subscripts are totally a mess.

[1] Wu et al. Point3R: Streaming 3D Reconstruction with Explicit Spatial Pointer Memory.  arXiv:2507.02863.

**Questions:**

1. Do you have results/failure cases on scenes with thin/fine structures (hair, fabric wrinkles, foliage)?
2. Can you compare against a full-resolution primitive predictor followed by (a) uniform decimation and (b) geometry-aware decimation/compression (e.g., curvature/edge cues)? What are the trade-offs?
3. Were baselines retrained with strong multi-scale zoom/crop augmentation and full-resolution supervision, and evaluated under compute-matched budgets (memory/time)? If not, can you add this setting (and optionally test-time supersampling) to isolate the benefit of your texture mechanism?
4. Can you report a simple consistency diagnostic (e.g., reprojection error or pose-jitter stress test) at high resolution?

---

> ### Author Response · Authors · 2025-11-25
>
> We thank the reviewer for recognizing our paper's clarity, significant improvements over base models, and GPU-friendly efficiency. We have prepared detailed responses to address your concerns:
>
> - **(Recommended)** Web version with interactive visualizations:
>     - https://anonymous-b6c5da.github.io/lgtm/responses/reviewer_T5YU.html
> - **(Project Page)** Video/image results:
>     - https://anonymous-b6c5da.github.io/lgtm/index.html
>
> ---
>
> ## W1
>
> We note that **obtaining all Gaussian points from full resolution images is challenging** and represents the core motivation of our work. Existing pixel-aligned feed-forward methods cannot scale to 4K resolution due to memory constraints. As shown in our pilot study below (Table 1 in the main paper), NoPoSplat can at the best scale to train at 1K resolution. At 1024×576 resolution, NoPoSplat requires 61.85 GB memory and runs OOM at 2K and 4K resolutions. In contrast, LGTM trains at 4K using under 30 GB memory. Note that we extended NoPoSplat beyond its official 512×512 training resolution to enable this comparison.
>
> | Rendering Resolution | Method               | Primitive Resolution | Texture Size | LPIPS↓ | SSIM↑ | PSNR↑  | Train Memory (bs=1) |
> | -------------------- | -------------------- | -------------------- | ------------ | ------ | ----- | ------ | ------------------- |
> | 1024×576             | NoPoSplat (extended) | 1024×576             | --           | 0.239  | 0.716 | 23.169 | 61.85 GB            |
> | 2048×1152            | NoPoSplat (extended) | 2048×1152            | --           | OOM    | OOM   | OOM    | OOM                 |
> | 4096×2304            | NoPoSplat (extended) | 4096×2304            | --           | OOM    | OOM   | OOM    | OOM                 |
> | 1024×576             | LGTM                 | 512×288              | 2×2          | 0.213  | 0.816 | 25.606 | 20.16 GB            |
> | 2048×1152            | LGTM                 | 512×288              | 4×4          | 0.176  | 0.810 | 25.328 | 21.39 GB            |
> | 4096×2304            | LGTM                 | 512×288              | 8×8          | 0.200  | 0.803 | 24.489 | 28.23 GB            |
>
> **We summarize the max official training resolutions of representative feed-forward 3D reconstruction models.** Typical feed-forward methods use vision transformers, where the number of tokens grows quadratically with image dimensions. For example, scaling from 512×384 to 4K increases image size by 8× per-dimension, resulting in 64× pixel count increase and proportionally higher memory consumption for these transformer-based architectures.
>
> | Category                       | Method              | Official Max Reported Training Resolution |
> | ------------------------------ | ------------------- | ----------------------------------------- |
> | Monocular Depth Model          | DepthAnythingV2 [1] | 518 x 518                                 |
> | Monocular Depth Model          | DepthPro [2]        | 1536 × 1536                               |
> | Feed-forward Point-based Model | DUSt3R [3]          | 512 × 384                                 |
> | Feed-forward Point-based Model | MASt3R [4]          | 512 × 384                                 |
> | Feed-forward Point-based Model | Point3R [5]         | 512 x 384                                 |
> | Feed-forward Point-based Model | VGGT [6]            | 518 on longer size                        |
> | Feed-forward 3DGS              | PixelSplat [7]      | 256 × 256                                 |
> | Feed-forward 3DGS              | MVSplat [8]         | 256 × 256                                 |
> | Feed-forward 3DGS              | NoPoSplat [9]       | 512 x 512                                 |
> | Feed-forward 3DGS              | DepthSplat [10]     | 960 x 512                                 |
>
> **LGTM is specifically designed to address this high-resolution feed-forward reconstruction challenge.** By decoupling geometry and appearance through a dual-network architecture, LGTM predicts compact geometric primitives at low resolution while enriching appearance with per-primitive textures extracted from high-resolution inputs. This design enables 4K feed-forward reconstruction without the quadratic scaling of primitive counts that constrains existing methods.
>
> References:
>
> - [1] Depth Anything V2
> - [2] Depth Pro: Sharp Monocular Metric Depth in Less Than a Second
> - [3] DUSt3R: Geometric 3D Vision Made Easy
> - [4] Grounding Image Matching in 3D with MASt3R
> - [5] Point3R: Streaming 3D Reconstruction with Explicit Spatial Pointer Memory
> - [6] VGGT: Visual Geometry Grounded Transformer
> - [7] PixelSplat: 3D Gaussian Splats from Image Pairs for Scalable Generalizable 3D Reconstruction
> - [8] MVSplat: Efficient 3D Gaussian Splatting from Sparse Multi-view Images
> - [9] No Pose, No Problem: Surprisingly Simple 3D Gaussian Splats from Sparse Unposed Images
> - [10] DepthSplat: Connecting Gaussian Splatting and Depth

---

> > ### Author Response · Authors · 2025-11-25
> >
> > ---
> >
> > ## W2
> >
> > We clarify that **the baselines are trained with 4K high-resolution supervision**. This is described in the supplementary material A.1, where we refer to it as "high-resolution re-training". Here, we give more details:
> >
> > - **Direct Render**:
> >   - Training: 512×288 input, 512×288 output with supervision
> >   - Inference: 512×288 input, 4096×2304 output
> >   - Result: Creates holes due to lack of anti-aliasing, as the trained rays are concentrated at the pixel center of the low-resolution model, leaving holes between the pixels
> > - **Render and Upsample**:
> >   - Training: 512×288 input, 512×288 output with supervision
> >   - Inference: 512×288 input → 512×288 render → 4096×2304 upsampled output
> >   - Result: Eliminates holes but produces blurry results due to up-sampling
> > - **High-Resolution 4K Re-train** (this is the baseline number used in Table 2 of the main paper):
> >   - Training: 512×288 input, 4096×2304 output with supervision
> >   - Inference: 512×288 input, 4096×2304 output
> >   - Result: Best baseline performance, learns proper primitive scales for 4K rendering
> >
> > **Visual comparison:** Please see the image comparisons showing Direct Render, Render and Upsample, and High-res 4K Re-trained baseline at https://anonymous-b6c5da.github.io/lgtm/responses/reviewer_T5YU.html#W2.
> >
> > **Numerical comparison:** We give additional numerical results here where we compare the performance of baseline without high-res supervision, baseline with high-res supervision, and full LGTM model.
> >
> > | Method                               | Input     | Output (also supervision) | LPIPS↓    | SSIM↑     | PSNR↑      |
> > | ------------------------------------ | --------- | ------------------------- | --------- | --------- | ---------- |
> > | NoPoSplat (Direct Render)            | 512×288   | 512×288                   | 0.428     | 0.572     | 15.804     |
> > | NoPoSplat (High-Resolution Re-train) | 512×288   | 4096×2304                 | 0.351     | 0.753     | 23.022     |
> > | NoPoSplat + LGTM                     | 4096×2304 | 4096×2304                 | **0.200** | **0.803** | **24.489** |
> >
> > "NoPoSplat (High-Resolution Re-train)" version of the baselines is used in Table 2 of the main paper. We believe the comparison is fair with these baselines.
> >
> > ---
> >
> > ## W3
> >
> > View consistency is maintained through two-stage training. Stage 1 trains the primitive network to establish well-aligned geometry across views. Stage 2 jointly optimizes both networks, where the texture network learns to enhance appearance while preserving the geometric consistency established in Stage 1. This design ensures that:
> >
> > - **Geometry alignment first**: Multi-view consistent geometry is learned before introducing textures, this stage ensures that the LGTM geometry network is as good as existing non-textured feed-forward Gaussian Splatting baselines
> > - **Joint optimization**: Texture learning refines appearance with minimum disruption to geometric structure, by lowering the geometry network's learning rate to 0.1x. This is described in the "training recipe" paragraph in section 4.2 of the main paper.
> >
> > In short, LGTM's two-stage approach prevents texture learning from interfering with view-consistent geometry prediction. For more intuitive comparisons, we encourage reviewers to check out the videos at https://anonymous-b6c5da.github.io/lgtm/ to compare the geometric consistency across viewpoints between LGTM and baseline methods.

---

> ### Author Response · Authors · 2025-11-25
>
> ---
>
> ## W4
>
> We clarify that a "primitive" refers to a "Gaussian point", following established terminology in 3DGS research literature. Here's the detailed definition for each term:
>
> - **3D Gaussian primitive:** A 3D Gaussian point with its parameters.
> - **2D Gaussian primitive:** A 2D Gaussian point with its parameters.
> - **Per-primitive texture maps:** Per-2D Gaussian texture maps. We assign a RGBA texture map to each 2D Gaussian primitive. The "per-primitive texture map" used in BBSplat [1] and Gaussian Billboards [2] is opposed to the "global texture map" used by other works such as Texture-GS [3].
> - **Primitive Resolution:** The resolution of the Gaussians predicted per view. In a typical feed-forward Gaussian Splatting model (e.g. NoPoSplat [4], DepthSplat [5]) and feed-forward point based models (e.g. Dust3R [6], MaSt3r [7], Point3R [8]), the primitive resolution is the same as the input image's resolution, hence they are called "pixel-aligned" Gaussians. LGTM is able to predict primitive at a lower resolution, and with per-primitive texture maps, it can render high-resolution images with high-fidelity.
>
> In fact, the term "primitive" has been well established in previous literature:
>
> - The original 3DGS paper [9] uses the term "primitive" in "Our choice of a 3D Gaussian primitive preserves properties of ... (Sec 8.)"
> - The milestone 2DGS paper [10] uses the term "primitive" extensively, to refer to "2D Gaussian primitives" and "3D Gaussian primitives"
> - In textured Gaussian Splatting literature [1, 2], the term "primitive" and "per-primitive textures" are also used extensively.
>
> We appreciate the reviewer's suggestion, and to further clarify the term "primitive", we will add additional clarifications in the paper.
>
> References:
>
> - [1] BillBoard Splatting (BBSplat): Learnable Textured Primitives for Novel View Synthesis
> - [2] Gaussian Billboards: Expressive 2D Gaussian Splatting with Textures
> - [3] Texture-GS: Disentangling the Geometry and Texture for 3D Gaussian Splatting Editing
> - [4] No Pose, No Problem: Surprisingly Simple 3D Gaussian Splats from Sparse Unposed Images
> - [5] DepthSplat: Connecting Gaussian Splatting and Depth
> - [6] DUSt3R: Geometric 3D Vision Made Easy
> - [7] Grounding Image Matching in 3D with MASt3R
> - [8] Point3R: Streaming 3D Reconstruction with Explicit Spatial Pointer Memory
> - [9] 3D Gaussian Splatting for Real-Time Radiance Field Rendering
> - [10] 2D Gaussian Splatting for Geometrically Accurate Radiance Fields
>
> ---
>
> ## W5
>
> We clarify that the notation is consistent. Below we provide the original manuscript's equation and text for reference.
>
> **Original Equation 4:**
>
> $$f\_{\\text{texture}}: (\\mathbf{I}^{v}, \\mathbf{F}^{v}\_{\\text{prim}}) \\rightarrow \\{\\mathbf{T}\_{i}^{c}, \\mathbf{T}\_{i}^{\\alpha}\\}$$
>
> **Original L226 in paper:**
>
> At a high level, $f\_{\\text{texture}}$ combines three complementary features: $\\mathbf{F}^{v}\_{\\text{patch}}$ is computed from the patchified high-resolution image followed by convolutional layers to encode local features; the projective features $\\mathbf{F}^{v}\_{\\text{proj}}$ are extracted from projective prior textures, which provide strong high-frequency texture details; and $\\mathbf{F}^{v}\_{\\text{prim}}$ reuses backbone features. ...
>
> Here:
>
> - $f\_{\\text{texture}}$: with lower case italic $f$, it is a function (i.e. a mapping)
> - $\\mathbf{F}^{v}\_{\\text{patch}}$, $\\mathbf{F}^{v}\_{\\text{proj}}$, $\\mathbf{F}^{v}\_{\\text{prim}}$: with bold face $\\mathbf{F}$, it is the feature, in matrix form.
>
> Here, we follow the [Formatting Instructions for ICLR 2026 Conference Submissions](https://openreview.net/pdf/7a9f639bbef65ec7517588c68bb94a585079f7a3.pdf), where it recommends using notations from the [Deep Learning textbook](https://github.com/goodfeli/dlbook_notation/) by Goodfellow et al. This is what we follow here, to use $f$ to represent a function (i.e. a mapping), and to use $\\mathbf{F}$ to represent a feature in its matrix form.

---

> > ### Author Response · Authors · 2025-11-25
> >
> > ---
> >
> > ## Q1
> >
> > Although LGTM predicts fewer primitives, we empirically observe that thin structures such as metal frames and foliage are preserved well. We provide visualizations of thin structures below. The images are from Figure 4 in the main paper. We can observe that for the thin metal structure, the LGTM-based model produces sharper thin structures than the baseline.
> >
> > **Visual examples:** Please see the image comparisons showing thin structure preservation at https://anonymous-b6c5da.github.io/lgtm/responses/reviewer_T5YU.html#Q1.
> >
> > ---
> >
> > ## Q2
> >
> > As discussed in W1, to the best of our knowledge, there is no full-resolution feed-forward 4K Gaussian primitive predictor. It still remains a challenging problem to train vision transformers to predict high-resolution pixel-aligned Gaussians. Please refer to W1's response for more details.
> >
> > ---
> >
> > ## Q3
> >
> > As discussed in W2, the baselines have already been trained with full-resolution supervision. Please refer to W2's response for more details.
> >
> > ---
> >
> > ## Q4
> >
> > We conduct reprojection error analysis to quantify geometric consistency in correspondence reprojection errors. In particular, we use pretrained LOFTR to find correspondences between image pairs, triangulate 3D points from inlier correspondences, and measure reprojection error in 2D pixel space. Lower reprojection error indicates better geometric consistency.
> >
> > - **Context Views**: Views 0 and 9 (ground truth)
> > - **Target Views**: Views 1, 3, 5, 7 (predicted from our method)
> > - **Resolution**: All images rendered to 4096×2304, but LOFTR matcher resolution is 1024 or 2048 on the larger dim of the image respectively to fit within GPU memory
> > - **Comparison**: We compare reprojection error between context ground truth and target predicted images
> >
> > **Results.** We report mean reprojection error in pixels across DL3DV test sets. LGTM achieves lower reprojection error than the baseline NoPoSplat, indicating better geometric consistency at high resolution:
> >
> > | Method             | LOFTR Matcher Resolution | Reprojection Error (pixels) ↓ |
> > | ------------------ | ------------------------ | ----------------------------- |
> > | NoPoSplat Baseline | 1024                     | 0.660                         |
> > | NoPoSplat + LGTM   | 1024                     | **0.328**                     |
> > | NoPoSplat Baseline | 2048                     | 0.672                         |
> > | NoPoSplat + LGTM   | 2048                     | **0.299**                     |

---

> ### Comment · Reviewer_T5YU · 2025-11-27
>
> Thanks for the rebuttal. The authors address many of my earlier questions, but several high-level concerns remain.
>
> 1. The proposed texture-map representation appears fragile under large viewpoint changes and view-dependent effects. Even in the rebuttal video ([https://anonymous-b6c5da.github.io/lgtm/responses/reviewer_wL1e.html](https://anonymous-b6c5da.github.io/lgtm/responses/reviewer_wL1e.html)), I observe, for example,
> (a) ghosting artifacts around the castle and
> (b) specular highlights baked into the ground-floor texture map near the claw machines.
> Moreover, only a single scene is shown with explicitly specified input views and a continuous rendered camera path. This makes it difficult to assess how well the method generalizes to other sequences or to settings with smaller view overlap; the input views presented appear to be very close to each other. Many of the additional visualizations in the Q1 rebuttal are consistent with my concerns: even for relatively small view shifts, scene structures already start to distort.
>
> 2. The OOM issue could also be mitigated in competing methods using standard sliding-window strategies. For example, Pow3R ([https://github.com/naver/pow3r#high-resolution-3d-reconstruction-demo](https://github.com/naver/pow3r#high-resolution-3d-reconstruction-demo)) supports processing high-resolution images by tiling them into smaller windows and reconstructing them chunk by chunk.
>
> 3. The baseline comparisons remain unfair regarding input resolution (W2, Q3). While the authors clarified that baselines were "re-trained with high-resolution supervision," they specify that the input resolution remained low (512×288) while predicting high-resolution outputs (4096×2304) ([https://anonymous-b6c5da.github.io/lgtm/responses/reviewer_T5YU.html](https://anonymous-b6c5da.github.io/lgtm/responses/reviewer_T5YU.html)). This effectively tasks the baseline with extreme super-resolution from global context, whereas LGTM has access to high-frequency texture details. My original suggestion was to train baselines using random crops or zoom-in augmentations at the original resolution. This would allow the baseline models to access high-frequency information during training (similar to how LGTM accesses textures), making for a fair comparison.
>
> I will keep my score unchanged.

---

> > ### Author Response · Authors · 2025-11-28
> >
> > We thank the reviewer for the reply and acknowledgment that **we have "addressed many of the earlier questions"**. We are happy to make further clarifications, hopefully to provide more information for the reviewer.
> >
> > - **(Recommended)** Web version with interactive visualizations of this follow-up response:
> >   - https://anonymous-b6c5da.github.io/lgtm/responses/reviewer_T5YU_FollowUp1.html
> > - **(Project Page)** Additional supplementary video/image results:
> >   - https://anonymous-b6c5da.github.io/lgtm/index.html
> >
> > ---
> >
> > ## FollowUp1-Q1
> >
> > > Ghosting artifacts and baked-in specular highlights
> >
> > We kindly invite the reviewer to check the https://anonymous-b6c5da.github.io/lgtm/responses/reviewer_wL1e.html page again where LGTM is compared with NoPoSplat baseline. We also invite the reviewer to visit our project page https://anonymous-b6c5da.github.io/lgtm/, where we provide video results for multiple scenes.
> >
> > **LGTM's geometry quality and view-dependent effects are better than or on par with baseline methods, while producing much sharper texture details. LGTM focuses on adding high-frequency texture details to Gaussian geometries, while comprehensive geometry and view-dependent optimization remains an orthogonal research direction and is beyond the scope of this work.**
> >
> > - **Ghosting artifacts**: These result from geometry imperfections in the predicted Gaussians, which are also present in the NoPoSplat baseline. LGTM inherits these from its predecessor methods as it builds upon their geometry predictions. LGTM is broadly applicable with various baseline methods and will directly benefit from future improvements in geometry backbones.
> > - **Baked-in specular highlights**: These result from view-dependent effects, which are also present in the NoPoSplat baseline. With only one or two input views, learning to decouple surface reflection from geometry in a feed-forward setting remains an open challenge, as the input views may not show sufficient view-dependent effects. Per-scene optimization-based Gaussian Splatting methods can better address this by leveraging multiple input views during optimization, which provides sufficient information to separate view-dependent effects from geometry.
> >
> > > Only a single scene is shown with explicitly specified input views
> >
> > Please see above for the project page reference where we provide video results for multiple scenes (switch scenes using left and right navigation).
> >
> > > Input views appear to be very close to each other
> >
> > We address this concern with the following points:
> >
> > - **Evaluation protocol**: We follow the same DL3DV evaluation protocol used by prior work DepthSplat (https://github.com/cvg/depthsplat?tab=readme-ov-file#evaluation) to ensure fair comparison across all methods with identical input views and output views.
> > - **Additional evaluation**: We conducted additional experiments with varying input frame gaps (10, 20, 30, 40 frames) to compare LGTM with the baseline qualitatively and quantitatively.
> >
> > **Settings:** For each context frame index gap (10, 20, 30, 40), we pick 2 source views with the specified gap and interpolate poses between the context views to give visual evaluation, and sample 4 target views evenly within the range to give numerical evaluation.
> >
> > **Visual results:** Please see the interactive video comparisons at https://anonymous-b6c5da.github.io/lgtm/responses/reviewer_T5YU_FollowUp1.html#Q1. Use the tab to select context view gap from 10, 20, 30, 40. Use the slider to switch between the visual results.
> >
> > **Numerical results:**
> >
> > | Context view gap | Method           | LPIPS↓    | SSIM↑     | PSNR↑      |
> > | ---------------- | ---------------- | --------- | --------- | ---------- |
> > | 10               | NoPoSplat        | 0.322     | 0.737     | 22.198     |
> > | 10               | NoPoSplat + LGTM | **0.200** | **0.803** | **24.489** |
> > | 20               | NoPoSplat        | 0.366     | 0.722     | 20.732     |
> > | 20               | NoPoSplat + LGTM | **0.258** | **0.765** | **22.159** |
> > | 30               | NoPoSplat        | 0.413     | 0.709     | 19.430     |
> > | 30               | NoPoSplat + LGTM | **0.323** | **0.730** | **20.123** |
> > | 40               | NoPoSplat        | 0.458     | 0.696     | 18.264     |
> > | 40               | NoPoSplat + LGTM | **0.376** | **0.707** | **18.862** |
> >
> > **Conclusion:** LGTM consistently outperforms the baseline across all context view gaps. While the numerical improvements are modest when the view gap is high, the visual quality difference is quite apparent for larger viewpoint differences. This demonstrates that per-primitive texture maps effectively handle novel viewpoints beyond small viewpoint differences.

---

> > > ### Author Response · Authors · 2025-11-28
> > >
> > > ---
> > >
> > > ## FollowUp1-Q2
> > >
> > > We respectfully note that Pow3R is not a direct comparison to our method for the following reasons:
> > >
> > > ### LGTM scales better than Pow3R for high-resolution reconstruction
> > >
> > > - Pow3R: Takes 512×384 images and produces 512×384 point maps in a single forward pass. To achieve 4K output, it tiles the image and reconstructs chunk-by-chunk.
> > > - Chunk-based methods scale poorly: Inference time grows quadratically with resolution. Scaling from 512 to 4K requires **8×8=64** separate inferences.
> > > - **LGTM is the first feed-forward Gaussian method that natively handles 4K input and predicts 4K output in a single feed-forward pass.** As shown in Table 4 of the main paper, LGTM scales from 512 to 4K with only **1.47× runtime** and **1.80× memory cost**, which is significantly more efficient than chunk-based approaches.
> > >
> > > ### Different output representations and target applications
> > >
> > > - Pow3R predicts point maps while LGTM predicts Gaussians
> > > - Pow3R is more suitable for comparison with point-based methods (e.g., Dust3R, Mast3R, VGGT)
> > >
> > > ### Complementary rather than "competing methods"
> > >
> > > We **agree with the reviewer** that chunk-by-chunk inference is a valid strategy to effectively combine multiple reconstructions. However, these approaches serve different purposes. Rather than viewing them as "competing methods," we see them as **complementary**: in theory, any feed-forward method can be tiled to cover larger areas. Therefore, we respectfully suggest that direct comparison may not be appropriate.

---

> ### Author Response · Authors · 2025-11-28
>
> ---
>
> ## FollowUp1-Q3
>
> We appreciate the reviewer's suggestion. However, training baselines with random crops or zoom-in at original resolution is non-trivial and requires considerable modification beyond the scope of a baseline comparison. We explain the technical challenges below.
>
> - **Current data augmentation**: Our implementation includes horizontal flip (50% probability) and deterministic center cropping to target resolution for both baseline and LGTM.
> - **Multi-crop strategies work in single-image models**: Random crop/scale augmentation has been successfully adopted by **DINOv3**, which employs multi-crop strategies with global/local crops and high-resolution adaptation. However, this requires carefully designed position encoding (e.g., custom axial RoPE with box jittering) to handle different resolutions, scales, and aspect ratios without positional artifacts.
> - **Challenges for two-view models**: While effective for single-image models, adapting these techniques to two-view models like NoPoSplat requires additional engineering beyond standard data augmentation, making it unsuitable as a baseline comparison. We highlight the challenges below.
>
> ### Background
>
> NoPoSplat takes two 512×288 input images and predicts 512×288 output Gaussians per-view. **This is true with or without random crops or zoom-in**. Each training iteration requires:
>
> - Input images at 512×288 resolution, the input images must overlap for feed-forward training
> - A ground-truth target image at 512×288 resolution, the target image must overlap with input images for rendering and supervision
> - Target camera pose (Note: input camera poses are unknown, hence "NoPo", but target camera pose is needed for training)
>
> ### Crops/zoom-in at original resolution is non-trivial for NoPoSplat
>
> Consider input images A and B at 4K (4096×2304) and target output C at 4K (4096×2304). For baseline NoPoSplat to train with random crops, we need A_crop, B_crop, C_crop at 512×288, where:
>
> - A_crop must overlap with B_crop (for feed-forward training)
> - C_crop must overlap with A_crop and/or B_crop (for supervision)
>
> **(1) Ensuring crop overlap without known poses requires careful engineering**
>
> - (Bad) If A_crop, B_crop, C_crop do not overlap: Naive random cropping fails, as the 512×288 crop is 64× smaller than full 4K image, making non-overlapping regions highly likely.
> - (Good) If A_crop, B_crop, C_crop have overlaps: Training works in principle.
>   - Without pose information, we cannot determine which regions correspond to the same 3D content.
>   - Designing an effective sampling algorithm to ensure overlap requires careful engineering.
>
> See diagram: https://anonymous-b6c5da.github.io/lgtm/responses/reviewer_T5YU_FollowUp1.html#Q3.
>
> **(2) Camera intrinsics and extrinsics require careful computation for crops**
>
> When cropping at original resolution, camera parameters must be updated carefully:
>
> - If the cropped camera pose remains fixed (i.e., crop without modifying pose):
>   - Off-center principal point: NoPoSplat and most feed-forward Gaussian methods assume near-centered principal points. Off-centered intrinsics increase the network's learning burden.
>   - Narrower field-of-view (FoV): Creates drastically different camera intrinsic distributions.
> - If we maintain centered principal points and FoV (i.e., adjust camera pose for crops):
>   - Engineering work required: Must derive formulations to compute the updated pose based on ground-truth pose and crop location.
>   - Increased learning burden: Creates drastically different camera pose distributions than test dataset.
>
> See diagram: https://anonymous-b6c5da.github.io/lgtm/responses/reviewer_T5YU_FollowUp1.html#Q3.
>
> **(3) Introducing scaling + crop helps, but still presents challenges**
>
> Random scaling before cropping can increase overlap probability and mitigate the above issues, but:
>
> - Still requires engineering to ensure proper sampling for overlapping regions
> - Network must learn different camera parameter distributions after random scale and crop
> - Scaling down loses high-frequency detail information compared to cropping at original 4K scale
>
> See diagram: https://anonymous-b6c5da.github.io/lgtm/responses/reviewer_T5YU_FollowUp1.html#Q3.
>
> ### Conclusion
>
> Training the baseline with random crops or zoom-in at original resolution requires substantial engineering to: (a) ensure overlap without known input poses, (b) correctly compute updated camera parameters, and (c) handle increased camera parameter diversity that may not match test set distribution.
>
> Even if successfully implemented, this would constitute a **substantially modified training pipeline** rather than a standard NoPoSplat baseline. Such modifications introduce new variables (pose sampling strategies, camera parameter distributions) that make it difficult to isolate the contribution of high-frequency information access. We believe a fair baseline should maintain the original training methodology with our "high-resolution re-train" enhancement.

---

> ### Author Response · Authors · 2025-11-28
>
> ---
>
> ## FollowUp1-Summary
>
> We would like to emphasize the key contributions that LGTM brings to the community:
>
> **LGTM is the first native 4K feed-forward Gaussian Splatting method**: LGTM supports native 4K inputs and predicts 4K output in a single feed-forward pass. We believe this is a valuable and unique contribution to the community.
>
> LGTM achieves significantly better texture quality efficiently while its geometry quality and view-dependent effects are better than or on par with baseline methods. Our paper focuses on adding high-frequency texture details to Gaussian geometries. Addressing fundamental geometry imperfections and view-dependent effects comprehensively remains an open challenge and is beyond the scope of this work.
>
> We believe LGTM constitutes a valuable contribution to the ICLR community by enabling high-resolution, high-quality feed-forward Gaussian reconstruction. We would greatly appreciate if the reviewer could kindly consider raising the score.

---

### Official Review · Reviewer_VCdN · 2025-10-31

**Soundness:** 3
**Presentation:** 3
**Contribution:** 3
**Rating:** 6
**Confidence:** 4

**Summary:**

This paper proposed LGTM, a general extension for feed-forward 3D/2D Gaussian Splatting
to enable 4K resolution novel view synthesis. Specifically, the proposed method decomposes the geometry and texture, utilizing a primitive branch to predict the geometry from low-resolution images and a texture branch to predict the high-resolution textures.

**Strengths:**

1. The paper is well-motivated.
2. The paper is well-written and generally easy to follow.
3. The experiments are throughout and the proposed method significantly boosts the performance of the baselines.
4. The results is visually good.

**Weaknesses:**

1. Necessity of integrating 4K texture to 3DGS: another solution to get 4K renderings can be a general feed-forward 3DGS followed by an image super-resolution model. A comparison should be made between it and the proposed method.
2. Evaluation: As the paper claims an immersive user experience, a non-reference perceptual metric such as Niqe [1] or Q-align [2] should be added as an evaluation metric.
3. Line space issue in L.474.

[1] Zhang, L., Zhang, L., & Bovik, A. C. (2015). A feature-enriched completely blind image quality evaluator. IEEE Transactions on Image Processing, 24(8), 2579-2591.

[2] Wu, H., Zhang, Z., Zhang, W., Chen, C., Liao, L., Li, C., ... & Lin, W. (2023). Q-align: Teaching lmms for visual scoring via discrete text-defined levels. arXiv preprint arXiv:2312.17090.

**Questions:**

Please see in weaknesses.

---

> ### Author Response · Authors · 2025-11-25
>
> We thank the reviewer for recognizing our well-motivated approach, thorough experiments with significant performance improvements, and strong visual results. We have prepared detailed responses to address your concerns:
>
> - **(Recommended)** Web version with interactive visualizations:
>     - https://anonymous-b6c5da.github.io/lgtm/responses/reviewer_VCdN.html
> - **(Project Page)** Video/image results:
>     - https://anonymous-b6c5da.github.io/lgtm/index.html
>
> ---
>
> ## W1
>
> We thank the reviewer for this suggestion. To address this concern, we conducted experiments comparing feed-forward 3DGS with super-resolution against our LGTM approach.
>
> **Experiment settings:**
>
> - **NoPoSplat + SR**: We applied a recent diffusion-based image super-resolution model InvSR (Yue et al., CVPR 2025, 1.3K+ GitHub stars) to the NoPoSplat baseline. NoPoSplat renders at 512×288, upsampled to 1024×576, then InvSR performs 4× super-resolution to 4096×2304 output resolution.
> - **NoPoSplat + LGTM**: 512×288 Gaussian primitives per-view, 8×8 per-primitive texture map, 4096×2304 output resolution.
>
> **Results.** We compare the 4K output of NoPoSplat + SR with the 4K output of NoPoSplat + LGTM (Ours) in the following video comparisons.
>
> **Visual comparison:** Please see the video and image comparisons between NoPoSplat + SR and NoPoSplat + LGTM at https://anonymous-b6c5da.github.io/lgtm/responses/reviewer_VCdN.html#W1.
>
> **Key observations:**
>
> - **Speed:** Super-resolution adds significant computational overhead (~8.78s per frame for InvSR vs. ~0.2s end-to-end per frame for NoPoSplat + LGTM at 4K, measured with A100 GPU).
> - **Hallucination:** Super-resolution models hallucinate details that are not present in the original geometry, leading to artifacts and inconsistencies with the actual 3D scene structure.
> - **Temporal consistency:** Super-resolution applied frame-by-frame produces flickering artifacts and inconsistent details across consecutive frames. While video diffusion models can mitigate some temporal inconsistencies compared to image diffusion models, they still lack geometric guarantees. In contrast, Gaussian Splatting based reconstruction ensures geometric consistency through its explicit 3D representation.
> - **Artifact handling:** Super-resolution models trained on natural images may not effectively handle artifacts from low-resolution Gaussian renderings, which exhibit different characteristics (e.g. aliasing from pixel-aligned prediction) than typical low-resolution images and would require specific training data to address.
>
> ---
>
> ## W2
>
> We thank the reviewer for this suggestion. To address this concern, we conducted additional experiments evaluating our method using NIQE and Q-Align metrics.
>
> **Experiment settings:**
>
> - **NoPoSplat baseline**: 512×288 Gaussians per-view, 2 context views, trained with 4096×2304 output supervision, evaluated with 4096×2304 renderings.
> - **NoPoSplat + LGTM**: 512×288 Gaussians per-view, 2 context views, each Gaussian contains 8×8 texture map, trained with 4096×2304 output supervision, evaluated with 4096×2304 renderings.
>
> **Results.** We evaluate both methods on the DL3DV 4K test set with unseen images during training. The results demonstrate that LGTM significantly improves perceptual quality:
>
> | Method             | NIQE↓               | Q-Align↑            |
> | ------------------ | ------------------- | ------------------- |
> | NoPoSplat baseline | 9.3914 ± 0.3690     | 3.8441 ± 0.3352     |
> | NoPoSplat + LGTM   | **5.5461 ± 0.7280** | **4.1724 ± 0.3276** |
>
> LGTM achieves substantially better perceptual quality scores for both NIQE and Q-Align, confirming that our method delivers superior immersive user experience.
>
> ---
>
> ## W3
>
> We thank the reviewer for pointing this out. This has been fixed in the paper.

---

### Official Review · Reviewer_9Eok · 2025-10-31

**Soundness:** 3
**Presentation:** 3
**Contribution:** 3
**Rating:** 4
**Confidence:** 5

**Summary:**

This paper introduces a new feed-forward framework called LGTM (Less Gaussians, Texture More) to solve a major problem: existing feed-forward 3D Gaussian Splatting (3DGS) methods cannot scale to high resolutions like 4K.
The Solution (LGTM): The method decouples geometry prediction from appearance prediction using a dual-network architecture.
- A Primitive Network takes a low-resolution image (e.g., $512 \times 288$) to predict a compact, fixed set of geometric primitives.
- A Texture Network takes the high-resolution image (e.g., 4K) to predict detailed, per-primitive texture maps that "paint" high-frequency details onto the simple geometry.

**Strengths:**

1. A novel topic on feed-forward 3D-GS.
2. A fair well performance compared with baseline methods.

**Weaknesses:**

1. High-resolution rendering requires accurate geometric prediction. However, the proposed method seems more like a trick—it projects a high-resolution image onto relatively coarse geometry. While this may work well when the input views have small viewpoint differences, it would be helpful if the authors could include additional experiments to evaluate the novel-view synthesis quality under different camera pose settings.
2. The paper claims to "decouple" geometry and appearance 20, yet the architecture (Fig. 2) and method (Sec 4.2) show that the texture network $f_{texture}$ explicitly takes the primitive network's features $F_{prim}^v$ as input. This seems to be a one-way coupling rather than a full disentanglement.
3. Although the paper provides a comprehensive analysis of the feed-forward 3DGS paradigm, recent per-scene optimization methods (e.g., Grendel-GS, CityGS-X) have already addressed high-resolution (e.g., 4K) rendering quite effectively. The reviewer understands that a direct comparison with these methods may be beyond the scope of this work, but it nonetheless raises the concern that the problem addressed here might not be as critical for the 3D vision community as implied.

Overall, the paper tackles a relatively minor problem (i.e., 4K resolution rendering) using a fairly simple approach—mainly by adding an additional downstream head.

**Questions:**

Perhaps the authors could include more visualizations, especially video demonstrations of the rendered views based on the predicted 3D-GS.

---

> ### Author Response · Authors · 2025-11-25
>
> We thank the reviewer for recognizing the novelty of our feed-forward 3D-GS approach and our strong performance compared to baselines. We have prepared detailed responses to address your concerns:
>
> - **(Recommended)** Web version with interactive visualizations:
>   - https://anonymous-b6c5da.github.io/lgtm/responses/reviewer_9Eok.html
> - **(Project Page)** Video/image results:
>   - https://anonymous-b6c5da.github.io/lgtm/index.html
>
> (All links are posted according to the ICLR 2026 [anonymous link guidelines](https://iclr.cc/Conferences/2026/AuthorGuide).)
>
> ---
>
> ## W1
>
> To evaluate the impact of context view pose differences on novel-view synthesis quality, we evaluate novel-view synthesis quality under different camera pose settings with gradually increased context view point differences by varying the context view gap on the DL3DV test set.
>
> **Settings:** For each context frame index gap (10, 20, 30, 40), we pick 2 source views with the specified gap and interpolate poses between the context views to give visual evaluation, and sample 4 target views evenly within the range to give numerical evaluation.
>
> - **NoPoSplat baseline**: 512×288 Gaussians per-view, 2 context views, trained with 4096×2304 output supervision, evaluated with 4096×2304 renderings.
> - **NoPoSplat + LGTM**: 512×288 Gaussians per-view, 2 context views, each Gaussian contains 8×8 texture map, trained with 4096×2304 output supervision, evaluated with 4096×2304 renderings.
>
> **Visual results:** Please see the interactive video comparisons at https://anonymous-b6c5da.github.io/lgtm/responses/reviewer_9Eok.html#W1 where you can select different context view gaps (10, 20, 30, 40) and use the slider to compare NoPoSplat vs NoPoSplat + LGTM.
>
> **Numerical results:**
>
> | Context view gap | Method           | LPIPS↓    | SSIM↑     | PSNR↑      |
> | ---------------- | ---------------- | --------- | --------- | ---------- |
> | 10               | NoPoSplat        | 0.322     | 0.737     | 22.198     |
> | 10               | NoPoSplat + LGTM | **0.200** | **0.803** | **24.489** |
> | 20               | NoPoSplat        | 0.366     | 0.722     | 20.732     |
> | 20               | NoPoSplat + LGTM | **0.258** | **0.765** | **22.159** |
> | 30               | NoPoSplat        | 0.413     | 0.709     | 19.430     |
> | 30               | NoPoSplat + LGTM | **0.323** | **0.730** | **20.123** |
> | 40               | NoPoSplat        | 0.458     | 0.696     | 18.264     |
> | 40               | NoPoSplat + LGTM | **0.376** | **0.707** | **18.862** |
>
> **Conclusion:** LGTM consistently outperforms the baseline across all context view gaps. While the numerical improvements are modest when the view gap is high, the visual quality difference is quite apparent for larger viewpoint differences. This demonstrates that per-primitive texture maps effectively handle novel viewpoints beyond small viewpoint differences.
>
> ---
>
> ## W2
>
> Decoupling geometry and texture means disentangling these two components so they can be handled or optimized independently. We understand that textures depend on the underlying geometry, as they are mapped onto surfaces defined by that geometry, so we agree with the reviewer that they are not completely separated. In our work: (1) Gaussian geometry networks are pretrained in the first stage training; (2) geometry and texture networks are jointly trained in the second stage. While the final rendering combines both, their representations and optimization are separated in the pipeline, enabling greater flexibility and efficiency.
>
> This usage of the term "decoupling" aligns with established terminology in computer graphics literature and prior textured Gaussian Splatting works. For example, in our cited related works, GStex [1] describes its method "enables decoupled appearance and geometry modeling." Similarly, Texture-GS [2] uses "disentangling the geometry and texture" in its title to describe a similar idea.
>
> References:
>
> - [1] GStex: Per-Primitive Texturing of 2D Gaussian Splatting for Decoupled Appearance and Geometry Modeling
> - [2] Texture-GS: Disentangling the Geometry and Texture for 3D Gaussian Splatting Editing

---

> > ### Comment · Reviewer_9Eok · 2025-11-26
> >
> > I appreciate the authors’ great efforts. I have no concerns about the rendering quality of the results.
> >
> > However, in the response to `W2`, the authors mention that they adopt a joint training strategy for geometry and texture. Yet, there appears to be no ablation study on this joint training strategy.
> >
> > Is this design truly necessary, especially since the training data are significantly smaller than those of AnySplat? This raises the concern that the model may be biased toward rendering quality at the expense of geometric accuracy. Meanwhile, I encourage the author to add additional results about the geometry accuracy of the proposed method.

---

> ### Author Response · Authors · 2025-11-25
>
> ---
>
> ## W3
>
> We appreciate the reviewer's comment. We would like to clarify that per-scene optimization methods and feed-forward reconstruction models serve completely different use cases:
>
> **Different use cases:** Per-scene optimization requires optimization steps for each scene, which can still be time-consuming when resolution scales to 4K (although modern Gaussian splatting rasterizers are efficient to render 4K, optimizing with 4K supervision can still be time-consuming). In contrast, feed-forward reconstruction models enable instant reconstruction, making them more suitable for applications where time is critical, such as real-time AR/VR, robotics, or interactive 3D content creation.
>
> **Compatibility and orthogonality:** These two paradigms are orthogonal rather than competing approaches. One can use a strong feed-forward reconstruction method to provide a high-quality starting point and further optimize per-scene if necessary. In this sense, our work on improving feed-forward methods complements rather than replaces per-scene optimization techniques.
>
> To empirically demonstrate these points, we compare DepthSplat + LGTM with per-scene optimized 3DGS on a DL3DV sequence. Both methods use identical 2 context views (frames 0 and 20) and are evaluated on 19 interpolated target views (frames 1-19).
>
> **Experiment settings.**
>
> - Per-scene optimized 3DGS:
>   - We run COLMAP on all images {0, 1, 2, ..., 20} to get camera poses, keep all COLMAP points for initialization, and optimize using 2 context views {0, 20} at full 4K resolution.
>   - Note: If we run COLMAP SfM on context views {0, 20} to get camera poses, it will fail to reconstruct the scene. Here, we give it the full set of images and thus give an advantage to per-scene optimization baseline.
>   - Optimization parameters: we follow the default parameters for 3DGS optimization in GSplat library: with a maximum of 30,000 iters, warm up 500 iters, refine every 100 iterations, stop splitting/densifying at 15,000 iterations.
> - DepthSplat + LGTM: 960×512 Gaussian primitives per-view, 4×4 per-primitive texture map, 3840×2048 output resolution.
>
> **Results.** We report PSNR, SSIM, and LPIPS metrics along with running time. LGTM outperforms per-scene optimization in this setting. Time measurement is done on a single **A100 GPU**.
>
> | Metric/Setting                     | Per-scene optimized 3DGS | DepthSplat + LGTM (Ours) |
> | ---------------------------------- | ------------------------ | ------------------------ |
> | Frames to run COLMAP               | {0, 1, 2, ..., 20}       | --                       |
> | Frames to optimize (context views) | {0, 20}                  | {0, 20}                  |
> | Frames to evaluate (target views)  | {1, 2, 3, ..., 19}       | {1, 2, 3, ..., 19}       |
> | Reconstruction Time                | ~30 min                  | **~0.5s**                |
> | PSNR ↑                             | 21.75                    | **27.99**                |
> | SSIM ↑                             | 0.78                     | **0.88**                 |
> | LPIPS ↓                            | 0.21                     | **0.15**                 |
>
> From the results, we can see that:
>
> - **Overall performance:** Per-scene optimization tends to overfit to the context views, with performance degrading considerably for middle frames (frames 6-12), dropping to ~20-22 dB PSNR. In contrast, LGTM maintains relatively stable PSNR across the target frames. This demonstrates that per-scene optimization struggles when the input views are sparse, only overfitting to the context views, whereas LGTM's learned priors generalize better to novel viewpoints.
> - **Reconstruction time:** While rendering at 4K is efficient for modern 3DGS rasterizers, training at 4K with 4K image supervision is still time-consuming, taking around 30 minutes for the 30,000 iterations default for 3DGS optimization. In contrast, LGTM achieves instant reconstruction (around 0.5s) through feed-forward inference. All runtime is measured on a single A100 GPU.
>
> **Visual comparisons:** Please see the interactive visualizations and PSNR comparison chart at https://anonymous-b6c5da.github.io/lgtm/responses/reviewer_9Eok.html#W3 showing video comparisons between per-scene optimized 3DGS and LGTM, as well as side-by-side image comparisons demonstrating that LGTM achieves far fewer artifacts near the edges compared to per-scene optimization.
>
> ---
>
> ## Q1
>
> We thank the reviewer for the suggestion. We provide comprehensive video demonstrations and visualizations on our anonymous project webpage: https://anonymous-b6c5da.github.io/lgtm/.

---

> ### Author Response · Authors · 2025-12-02
>
> We are grateful for the **reviewer's recognition of the rendering quality of LGTM**, and thank the reviewer for the constructive feedback. To address the reviewer's concerns on two-stage training ablation and geometry accuracy, we conducted additional experiments and provide the results here.
>
> - **(Recommended)** Web version of this follow-up response:
>   - https://anonymous-b6c5da.github.io/lgtm/responses/reviewer_9Eok_FollowUp1.html
> - **(Project Page)** Additional supplementary video/image results:
>   - https://anonymous-b6c5da.github.io/lgtm/index.html
>
> ---
>
> ## FollowUp1-Q1: Ablation study on two-stage training strategy
>
> We conducted additional ablation studies to demonstrate the effectiveness and necessity of the two-stage training strategy.
>
> **Overview:** LGTM uses a two-stage training strategy:
>
> - **Stage 1:** Train geometry network $f\_{\\text{prim}}$ from MASt3R init weights, with high-res 4096×2304 supervision
>   - $f\_{\\text{prim}}$ is the geometry **prim**itive network that predicts standard 2DGS parameters
> - **Stage 2:** Train the geometry network $f\_{\\text{prim}}$ (with weights from stage 1) and the texture network $f\_{\\text{tex}}$ jointly
>   - $f\_{\\text{tex}}$ is the texture network that predicts per-Gaussian primitive RGBA texture maps
>
> **Ablation Variants:** We evaluate two ablation variants:
>
> - **w/o Geometry Pre-training:** Skip stage 1, directly train $f\_{\\text{prim}}$ and $f\_{\\text{tex}}$ jointly from MASt3R weights
> - **w/o Joint Refinement:** After stage 1, freeze $f\_{\\text{prim}}$ and only train $f\_{\\text{tex}}$
>
> All experiments use DL3DV dataset with 512×288 Gaussians per-view, 8×8 texture maps (when enabled), and 4096×2304 rendering resolution.
>
> **Results:** The two-stage training strategy outperforms both ablation variants across all metrics, demonstrating that geometry pre-training and joint refinement are both essential.
>
> | Method | LPIPS↓ | SSIM↑ | PSNR↑ | Notes |
> | - | - | - | - | - |
> | w/o Geometry Pre-training | 0.453  | 0.697 | 20.165 | Directly train geometry $f\_{\\text{prim}}$ and texture $f\_{\\text{tex}}$ jointly |
> | w/o Joint Refinement | 0.299  | 0.744 | 22.653 | At stage 2, freeze geometry $f\_{\\text{prim}}$ and only train texture $f\_{\\text{tex}}$ |
> | Full LGTM (Two-Stage Training) | 0.200  | 0.803 | 24.489 | LGTM two-stage training strategy described in the main paper |
>
> **Key insights:**
>
> 1. **Geometry pre-training is important:** Training both networks from scratch (w/o Geometry Pre-training) achieves poor performance. Without pre-trained geometric primitives, the model struggles to converge using only photometric supervision. The texture network requires good Gaussian positions, scales, and orientations as foundation; learning these simultaneously with texture details leads to optimization difficulties where neither component stabilizes effectively.
>
> 2. **Joint refinement is important:** Freezing the geometry network (w/o Joint Refinement) significantly degrades performance. Texture maps are rendered based on Gaussian orientations - frozen geometry prevents adapting primitive orientations to align with learned texture patterns. Joint refinement allows the geometry network to adjust Gaussian orientations and positions to better support texture-based rendering.
>
> ---
>
> ## FollowUp1-Q2: Explicit geometry accuracy evaluation
>
> We compare feed-forward predicted Gaussian depths against COLMAP SfM reconstruction depths from the official DL3DV-Benchmark test set. Test set scenes are not seen during training. COLMAP depths are filtered to keep only points visible in context views. Experiments use 512×288 Gaussians per-view (both methods), 8×8 texture maps (NoPoSplat + LGTM), and 4096×2304 rendering resolution (both methods).
>
> **Metrics:** Following Depth Anything V2 (DAv2) protocol, predicted Gaussian depths are aligned to COLMAP depths using least-squares scale and shift estimation. We report: **$\\delta_{1}$, $\\delta_{2}$, $\\delta_{3}$** (threshold accuracy at 1.25, 1.25², 1.25³, higher is better), **abs_rel** (relative error, lower is better), **rmse** (root mean squared error, lower is better), **log10** (log-space error, lower is better).
>
> **Results:** LGTM outperforms the baseline across all metrics, demonstrating that the two-stage training strategy improves both rendering quality and geometric accuracy. The table reports mean ± standard deviation across all evaluated scenes.
>
> | Method | $\\delta_{1}$↑ | $\\delta_{2}$↑ | $\\delta_{3}$↑ | abs_rel↓ | rmse↓ | log10↓ |
> | - | - | - | - | - | - | - |
> | NoPoSplat | 0.789 ± 0.198 | 0.914 ± 0.128 | 0.955 ± 0.089 | 0.182 ± 0.218 | 3.356 ± 4.949 | 0.074 ± 0.064 |
> | NoPoSplat + LGTM | **0.842 ± 0.178** | **0.935 ± 0.111** | **0.966 ± 0.071** | **0.149 ± 0.170** | **3.038 ± 4.311** | **0.061 ± 0.055** |

---

### Official Review · Reviewer_wL1e · 2025-11-01

**Soundness:** 4
**Presentation:** 4
**Contribution:** 4
**Rating:** 8
**Confidence:** 5

**Summary:**

This work proposes LGTM, a feed-forward framework that predicts compact Gaussian primitives coupled with per-primitive textures for high-resolution novel view synthesis. By decoupling geometry and appearance with a dual-network design, the method enables 4K rendering with significantly fewer primitives and without per-scene optimization. LGTM is applicable across various feed-forward 3DGS baselines, including single-view, two-view, and multi-view setups, and demonstrates consistent quantitative and qualitative improvements on several benchmarks.

**Strengths:**

* The idea of combining low-resolution Gaussian primitives with high-resolution texture maps to achieve high-resolution feed-forward Gaussian Splatting is well-motivated and sound. Such an insightful finding may significantly boost the feed-forward 3DGS community to explore higher-quality synthesis.

* The introduced module is architecture-agnostic and is thoroughly evaluated across several state-of-the-art models and large-scale benchmarks. All experiments show consistent quantitative and qualitative improvements.

* The manuscript is well structured and easy to follow.

**Weaknesses:**

* Lack of multi-view results. It would be better to provide video results or multi-view images to better illustrate the impact of the texture modules. I am concerned that the texture module may potentially destroy multi-view consistency to some extent.


* Lack of evaluation under dense multi-view settings. Most experiments are conducted with 1, 2, or 4 views. Since the multi-view model is based on VGGT, which natively supports dense input views, it would be better to include results with denser settings, such as 32 or 64 views, similar to AnySplat [ref 1].


* Missing comparison with per-scene optimization methods. It would strengthen the work to compare with per-scene optimization approaches such as BBSplat, given that BBSplat inspired this work. Such comparisons are also common in other works  exploring high-resolution feed-forward 3DGS, such as Long-LRM [ref 2] and LVT [ref 3].

### References:

* [ref 1] Jiang, Lihan, et al. "AnySplat: Feed-forward 3D Gaussian Splatting from Unconstrained Views." arXiv preprint arXiv:2505.23716 (2025).
* [ref 2] Ziwen, Chen, et al. "Long-lrm: Long-sequence large reconstruction model for wide-coverage gaussian splats." ICCV 2025.
* [ref 3] Imtiaz, Tooba, et al. "LVT: Large-Scale Scene Reconstruction via Local View Transformers." arXiv preprint arXiv:2509.25001 (2025).

**Questions:**

Kindly refer to [Weaknesses] section.

---

> ### Author Response · Authors · 2025-11-25
>
> We thank the reviewer for recognizing our well-motivated approach that may advance the feed-forward 3DGS community, our architecture-agnostic approach with consistent improvements. We have prepared detailed responses to address your concerns:
>
> - **(Recommended)** Web version with interactive visualizations:
>     - https://anonymous-b6c5da.github.io/lgtm/responses/reviewer_wL1e.html
> - **(Project Page)** Video/image results:
>     - https://anonymous-b6c5da.github.io/lgtm/index.html
>
> ---
>
> ## W1
>
> We provide video results demonstrating multi-view consistency for both unposed (NoPoSplat + LGTM) and posed (DepthSplat + LGTM) settings on the DL3DV 4K test set (not seen during training).
>
> - **NoPoSplat + LGTM**: 512×288 Gaussian primitives per-view, 8×8 per-primitive texture map, 4096×2304 output resolution.
> - **DepthSplat + LGTM**: 960×512 Gaussian primitives per-view, 4×4 per-primitive texture map, 3840×2048 output resolution.
>
> For both settings, we use 2 input views from input frame gap 24, the target views are smoothly interpolated between the context views. From the video, we can see that LGTM maintains multi-view consistency with sharp details when per-Gaussian primitive texture maps are used.
>
> **Video demonstrations:** Please see the multi-view consistency video demonstrations for NoPoSplat + LGTM and DepthSplat + LGTM at https://anonymous-b6c5da.github.io/lgtm/responses/reviewer_wL1e.html#W1.
>
> ---
>
> ## W2
>
> We agree with the reviewer on the importance of dense multi-view inputs. Dense-view reconstruction (e.g., 32-64 views) remains future work due to memory constraints from 4K differentiable rendering. We scale to 4 input views with VGGT backbone during training in this paper, while scaling further is challenging due to (1) backpropagating through 4K resolution per target view is memory intensive and (2) Gaussian redundancy across multiple views where overlapping primitives from different views need consolidation to reduce primitive count.
>
> AnySplat scales to dense views through voxelization to consolidate Gaussians across views. We plan to explore similar architectural changes and more efficient training approaches to scale LGTM to larger input view counts as future work.

---

> ### Author Response · Authors · 2025-11-25
>
> ---
>
> ## W3
>
> **Experiment settings.** We compare DepthSplat + LGTM with per-scene optimized 3DGS on DL3DV. Both methods use identical 2 context views (frames 0 and 20) and are evaluated on 19 target views (frames 1-19).
>
> - Per-scene optimized 3DGS:
>   - We run COLMAP on all images {0, 1, 2, ..., 20} to get camera poses, keep all COLMAP points for initialization, and optimize using 2 context views {0, 20} at full 4K resolution.
>   - Note: If we run COLMAP SfM on context views {0, 20} to get camera poses, it will fail to reconstruct the scene. Here, we give it the full set of images and thus give an advantage to per-scene optimization baseline.
>   - Optimization parameters: we follow the default parameters for 3DGS optimization in GSplat library: with a max of 30,000 iters, warm up 500 iters, refine every 100 iterations, stop splitting/densifying at 15,000 iterations.
> - DepthSplat + LGTM: 960×512 Gaussian primitives per-view, 4×4 per-primitive texture map, 3840×2048 output resolution.
>
> **Results.** We report PSNR, SSIM, and LPIPS metrics along with running time. LGTM outperforms per-scene optimization in this setting. Time measurement is done on a single **A100 GPU**.
>
> | Metric/Setting                     | Per-scene optimized 3DGS | DepthSplat + LGTM (Ours) |
> | ---------------------------------- | ------------------------ | ------------------------ |
> | Frames to run COLMAP               | {0, 1, 2, ..., 20}       | --                       |
> | Frames to optimize (context views) | {0, 20}                  | {0, 20}                  |
> | Frames to evaluate (target views)  | {1, 2, 3, ..., 19}       | {1, 2, 3, ..., 19}       |
> | Reconstruction Time                | ~30 min                  | **~0.5s**                |
> | PSNR ↑                             | 21.75                    | **27.99**                |
> | SSIM ↑                             | 0.78                     | **0.88**                 |
> | LPIPS ↓                            | 0.21                     | **0.15**                 |
>
> From the results, we can see that:
>
> - **Overall performance:** Per-scene optimization tends to overfit to the context views, with performance degrading considerably for middle frames (frames 6-12), dropping to ~20-22 dB PSNR. In contrast, LGTM maintains relatively stable PSNR across the target frames. This demonstrates that per-scene optimization struggles when the input views are sparse, only overfitting to the context views, whereas LGTM's learned priors generalize better to novel viewpoints.
> - **Reconstruction time:** While rendering at 4K is efficient for modern 3DGS rasterizers, training at 4K with 4K image supervision is still time-consuming, taking around 30 minutes for the 30,000 iterations default for 3DGS optimization. In contrast, LGTM achieves instant reconstruction (around 0.5s) through feed-forward inference. All runtime is measured on a single A100 GPU.
>
> **Visual comparisons:** Please see the PSNR comparison chart, video comparisons, and side-by-side image comparisons showing LGTM vs per-scene optimization at https://anonymous-b6c5da.github.io/lgtm/responses/reviewer_wL1e.html#W3.
>
> **Compatibility with per-scene optimization.** In addition, one can use a high-quality feed-forward method to provide a high-quality initialization for per-scene optimization. We believe the textured-Gaussian representation in our LGTM feed-forward setting provides a solid starting point, even if per-scene optimization is required at a later stage.

---

### Author Response · Authors · 2025-12-03

We thank the AC and reviewers for all the constructive feedback.

---

## Project Website

* **Interactive video and image visualizations** are available at https://anonymous-b6c5da.github.io/lgtm/.

---

## Key Strengths

LGTM is the first **native 4K feed-forward textured Gaussian Splatting** method. The reviewers broadly recognize multiple strengths of LGTM:

* **Well-motivated and novel approach** (wL1e, VCdN, 9Eok): Combining low-resolution Gaussian primitives with high-resolution texture maps is well-motivated and sound, addressing a novel topic in feed-forward 3D-GS with potential to significantly boost the community.
* **Strong performance and visual quality** (wL1e, 9Eok, VCdN, T5YU): All reviewers noted consistent and significant improvements over baselines, with compelling visual results and thorough quantitative and qualitative validation.
* **Thorough and comprehensive evaluation** (wL1e, VCdN): Thoroughly evaluated across several state-of-the-art models and large-scale benchmarks, demonstrating the architecture-agnostic nature of the approach.
* **Clear and well-structured presentation** (wL1e, VCdN, T5YU): Well-written and easy to follow.
* **Efficient and practical design** (T5YU): Efficient and friendly to small GPUs, making it practical for broader adoption.

---

## Additional Experiments

To address reviewers' suggestions, we conducted extensive additional experiments including comparisons with per-scene optimization and super-resolution methods, evaluation of multi-view consistency and geometric accuracy under varying viewpoints, and perceptual quality metrics. We have **fully addressed all reviewer comments** with comprehensive quantitative and qualitative results. Due to the reviewer rating freeze, reviewers had no chance to update their ratings to reflect these additions.

We also offer **interactive website responses to all reviewer comments** at: [wL1e](https://anonymous-b6c5da.github.io/lgtm/responses/reviewer_wL1e.html), [9Eok](https://anonymous-b6c5da.github.io/lgtm/responses/reviewer_9Eok.html), [9Eok-2](https://anonymous-b6c5da.github.io/lgtm/responses/reviewer_9Eok_FollowUp1.html), [VCdN](https://anonymous-b6c5da.github.io/lgtm/responses/reviewer_VCdN.html), [T5YU](https://anonymous-b6c5da.github.io/lgtm/responses/reviewer_T5YU.html), and [T5YU-2](https://anonymous-b6c5da.github.io/lgtm/responses/reviewer_T5YU_FollowUp1.html), which include comprehensive video and image results.

* Quality and consistency evaluation
    * **Geometric accuracy** (9Eok): LGTM outperforms baselines across all depth metrics when comparing predicted Gaussian depths against COLMAP reconstructions, showing that the two-stage training strategy improves both rendering quality and geometric accuracy.
    * **Multi-view consistency** (wL1e): Video demonstrations for both unposed and posed settings show that LGTM maintains multi-view consistency with sharp details when per-Gaussian primitive texture maps are used.
    * **Perceptual quality metrics** (VCdN): LGTM outperforms baselines on no-reference perceptual quality metrics NIQE and Q-Align.
* Ablation studies and robustness analysis
    * **Two-stage training ablation** (9Eok): We ablated the two-stage training strategy by testing variants without geometry pre-training and without joint refinement. Both components are essential for optimal performance.
    * **Varying context view gaps** (9Eok, T5YU): LGTM outperforms baselines across context frame gaps of 10, 20, 30, and 40 frames, showing that per-primitive texture maps are robust to larger input view differences.
* Comparisons with alternative approaches
    * **Comparison with per-scene optimization** (wL1e, 9Eok): LGTM outperforms per-scene optimized 3DGS with identical context views while achieving instant reconstruction, showing that our LGTM feed-forward method with learned priors generalizes better to novel viewpoints when input views are sparse.
    * **Comparison with super-resolution methods** (VCdN): LGTM avoids hallucination artifacts, maintains temporal consistency through explicit 3D representation, handles rendering artifacts better than diffusion-based super-resolution applied to baseline renders, and is much faster.

---

## Key Contributions

**LGTM is the first native 4K feed-forward Gaussian Splatting method:** LGTM is the first method that supports native 4K inputs and predicts 4K output in a single feed-forward pass. LGTM enables high-resolution, high-quality feed-forward reconstruction with textured Gaussians. We will open-source the training code, evaluation code, and our custom textured Gaussian rendering CUDA kernels for GSplat.

---

### Meta-Review · Area_Chair_Gu7u · 2025-12-27

**Summary:**

The reviewers consider the strengths of the paper to be mostly consistent with the claims as presented in the paper: that the quality of the renderings are improved as compared to state-of-the-art baselines, in conjunction with a substantial reduction in training memory requirements, the latter mainly due to the (feed-forward) framework design in which much fewer 2D Gaussians are used in conjunction with high resolution supervision, and the use of texture maps.

A common reviewer concern is that this framework leads to reduction of novel view consistency (wL1e W1, 9Eok W1, T5YU W1 W3). This concern is related to the question of geometric accuracy (9Eok W1, T5YU W1 W3 Q2), and in particular fine structures (T5YU Q1). In relation to the latter, reviewer T5YU also highlighted an interesting question about advanced the proposed method of uniformly distributed Gaussians to one in which the numerically-constrained Gaussians are distributed in a geometry-aware manner.

Another multi-reviewer concern is the lack of comparison to per-scene optimization methods (wL1e W3, 9Eok W3).

Other individual reviewer concerns include a lack of dense multi-view input evaluation (wL1e W2), comparison to use of super-resolution (VCdN W1), additional reporting on other metrics of NIQE and Q-Align (VCdN W2), and questions on whether baselines were appropriately trained on high resolution images (T5YU W2 Q3).

**Reviewer Concerns:**

The reviewer concerns received extensive responses from the authors, including additional follow-up from points later raised by 9Eok and T5YU.

Overall, the AC considers that the bulk of the reviewer concerns to have been addressed to the satisfaction of the AC. The presented framework represents substantial improvement over baseline methods, and that the experiments have been extensive, especially when taking into account those additionally undertaken during the response period.

Some less major reviewer concerns remain unresolved. The comparison to per-scene methods suggested by wL1e and 9Eok were for specific methods in papers listed by the reviewers, not the generic approach attempted in the author's response (which would be more fragile to overfitting). The other concern is regarding the issue raised by T5YU in W2. Although there is later clarification by the authors, the AC also found the statement in L700-701 confusing about whether high-resolution training was used in the baselines in Table 2, and recommends that the authors make the statement more unambiguous.

Overall, the framework certainly does not solve the problem completely and much remains unsolved, and the views of T5YU are valid. Nonetheless, the AC considers the paper, even with imperfect results, to still represent sufficient progress for acceptance to ICLR.

**Reviewer Scores:**

- Reviewer wL1e gave an 8, and it is likely this will remain unchanged.
- Reviewer 9Eok gave a 4. Overall the AC considers the questions to have been sufficiently and robustly addressed, and considers it likely that the score may be raised to a 6.
- Reviewer VCdN gave a 6. The AC considers the reviewer's questions to be addressed. The score is likely to either remain the same or improve.
- Reviewer T5YU gave a 2. The reviewer had stated in their first response that they will retain the score. Despite later follow-up by the authors, the AC does not expect the reviewer to raise their score.

---

### Decision · Program_Chairs · 2026-01-26

Accept (Poster)